EMBO
Molecular Medicine

# NrCAM is a marker for substrate-selective activation of ADAM10 in Alzheimer's disease

Tobias Brummer[1,2] ID, Stephan A Müller[1], Francisco Pan-Montojo[3,4] ID, Fumiaki Yoshida[5], Andreas Fellgiebel[6], Taisuke Tomita[5] ID, Kristina Endres[6] & Stefan F Lichtenthaler[1,2,3,7,*] ID

## Abstract

The metalloprotease ADAM10 is a drug target in Alzheimer's disease, where it cleaves the amyloid precursor protein (APP) and lowers amyloid-beta. Yet, ADAM10 has additional substrates, which may cause mechanism-based side effects upon therapeutic ADAM10 activation. However, they may also serve—in addition to APP—as biomarkers to monitor ADAM10 activity in patients and to develop APP-selective ADAM10 activators. Our study demonstrates that one such substrate is the neuronal cell adhesion protein NrCAM. ADAM10 controlled NrCAM surface levels and regulated neurite outgrowth *in vitro* in an NrCAM-dependent manner. However, ADAM10 cleavage of NrCAM, in contrast to APP, was not stimulated by the ADAM10 activator acitretin, suggesting that substrate-selective ADAM10 activation may be feasible. Indeed, a whole proteome analysis of human CSF from a phase II clinical trial showed that acitretin, which enhanced APP cleavage by ADAM10, spared most other ADAM10 substrates in brain, including NrCAM. Taken together, this study demonstrates an NrCAM-dependent function for ADAM10 in neurite outgrowth and reveals that a substrate-selective, therapeutic ADAM10 activation is possible and may be monitored with NrCAM.

**Keywords** acitretin; ADAM10; Alzheimer's disease; cerebrospinal fluid proteomics; NrCAM

**Subject Categories** Biomarkers & Diagnostic Imaging; Neuroscience

## Introduction

Alzheimer's disease (AD) is the most common neurodegenerative disorder and affects over 40 million patients worldwide (Scheltens *et al*, 2016; Selkoe & Hardy, 2016), but no causative or preventive treatment is currently available. One drug target in AD is the metalloprotease "a disintegrin and metalloprotease 10" (ADAM10), which acts as α-secretase and cleaves the amyloid precursor protein (APP) within the amyloid β (Aβ) domain (Lammich *et al*, 1999; Jorissen *et al*, 2010; Kuhn *et al*, 2010). Thus, ADAM10 cleavage of APP has the ability to prevent the generation of the pathogenic Aβ peptide. In fact, neuronal overexpression of wild-type ADAM10 in mouse brain prevents amyloid pathology whereas catalytically inactive ADAM10 suppresses α-secretase activity and consequently enhances amyloid pathology (Postina *et al*, 2004). Likewise, reduction of ADAM10 activity by interfering with ADAM10 trafficking enhanced amyloid pathology in mice (Epis *et al*, 2010). Importantly, rare, partial loss-of-function mutations of ADAM10 are genetically linked to late-onset AD in patients (Kim *et al*, 2009). This reinforces the notion that an activation of ADAM10 may be beneficial to treat or even prevent AD and has led to a first, phase 2 clinical trial with AD patients treated over 4 weeks with the ADAM10 activator acitretin (Endres *et al*, 2014). Acitretin is a second generation retinoid, is in clinical use to treat the skin disease psoriasis, and activates ADAM10 expression *in vitro* and in mice (Tippmann *et al*, 2009; Reinhardt *et al*, 2016). Importantly, in the small, proof-of-principle phase 2 study, acitretin indeed increased the ADAM10-mediated cleavage of APP (Endres *et al*, 2014). An in-depth neuropsychiatric examination was not performed in the patients and awaits a future study of longer duration and with a larger patient cohort. No major unwanted side effects were reported. Despite the positive outcome of the acitretin study, a key question is the safety profile of prolonged ADAM10 activation. This is of particular concern, because ADAM10 cleaves numerous other substrates besides APP—generally referred to as ectodomain shedding (Lichtenthaler *et al*, 2018)—, both during embryonic development and in different adult tissues, including the adult brain (Saftig & Lichtenthaler, 2015; Kuhn *et al*, 2016). One of its major substrates *in vivo* is the Notch receptor, which requires ADAM10 cleavage for its ligand-induced signal transduction (Pan & Rubin, 1997; Bozkulak & Weinmaster, 2009; van Tetering *et al*, 2009). Other substrates include cell adhesion

1  Deutsches Zentrum für Neurodegenerative Erkrankungen (DZNE), Munich, Germany
2  Neuroproteomics, School of Medicine, Klinikum rechts der Isar, Technische Universität München, Munich, Germany
3  Munich Cluster for Systems Neurology (SyNergy), Munich, Germany
4  Department of Neurology, Ludwig-Maximilians-University Munich, Munich, Germany
5  Laboratory of Neuropathology and Neuroscience, Graduate School of Pharmaceutical Sciences, The University of Tokyo, Tokyo, Japan
6  Department of Psychiatry and Psychotherapy, University Medical Center JGU, Mainz, Germany
7  Institute for Advanced Study, Technische Universität München, Garching, Germany
  *Corresponding author. Tel: +49 89 4400 46425; E-mail: stefan.lichtenthaler@dzne.de

proteins, e.g., NCAM and N-cadherin (Reiss *et al*, 2005; Hinkle *et al*, 2006), and growth factors and signaling proteins, such as neuregulin-1 (Freese *et al*, 2009), epidermal growth factor (Sahin *et al*, 2004), death receptor 6 (Colombo *et al*, 2018), and APLP2 (Endres *et al*, 2005; Hogl *et al*, 2011). Thus, the intended activation of ADAM10 for a prevention or treatment of AD may induce mechanism-based side effects by interfering with the cleavage and physiological function of other ADAM10 substrates. While this has not yet been tested systematically, a precedent is seen for γ-secretase inhibitors, which were discontinued in clinical trials for AD as they led to mechanism-based toxicity upon prolonged dosing (Golde *et al*, 2013). These inhibitors did not only block Aβ generation but also cleavage of additional γ-secretase substrates, including Notch.

Additional ADAM10 substrates are not only a concern, but also offer chances for drug development. Their cleavage products may be detected in body fluids, such as plasma and cerebrospinal fluid (CSF), and may potentially be used as companion diagnostics, i.e., as surrogate markers to monitor ADAM10 activity *in vivo*, similar to what has been suggested for the β-secretase BACE1 (Pigoni *et al*, 2016). Likewise, the additional substrates may be used for the development of substrate-selective ADAM10 activators that preferentially stimulate APP processing over the cleavage of other ADAM10 substrates.

Here, using CSF from a phase II clinical trial, we demonstrate that a substrate-selective activation of ADAM10 is feasible in patients and may be safer than expected. Moreover, we show that the ADAM10 substrate "neural glial-related cell adhesion molecule" (NrCAM) is an excellent marker for selective ADAM10 activation *in vivo*. NrCAM belongs to the L1 family of IgCAMs and is a cell adhesion molecule (Grumet *et al*, 1991), which controls dendritic spine densities, axonal guidance, and targeting as well as neurite outgrowth, by acting as a co-receptor molecule at the neuronal cell surface (Falk *et al*, 2004; Zelina *et al*, 2005; Nawabi *et al*, 2010; Torre *et al*, 2010; Demyanenko *et al*, 2011, 2014; Kuwajima *et al*, 2012; Dai *et al*, 2013). Soluble NrCAM (sNrCAM) is reduced in the CSF of AD patients compared to healthy controls (Hu *et al*, 2010; Wildsmith *et al*, 2014). In a previous proteomic study, NrCAM was discovered as an ADAM10 substrate *in vitro* (Kuhn *et al*, 2016). Now, we demonstrate that ADAM10 controls neuronal surface levels of NrCAM and neurite outgrowth in an NrCAM-dependent manner. Importantly, our study shows that activation of NrCAM cleavage by ADAM10 occurs through different mechanisms compared to APP, making soluble NrCAM an excellent marker for developing therapeutic, APP-selective, or APP-preferring ADAM10 activators.

## Results

### Proteolytic processing of NrCAM by furin, ADAM10, and γ-secretase

#### ADAM10 but not ADAM17 cleaves NrCAM in primary neurons

To determine whether the cleaved, soluble ectodomain of NrCAM (for a schematic, see Fig 1A) may be a suitable biomarker for ADAM10 activity, we first analyzed in detail the proteolytic processing of NrCAM in cultured neurons. The ADAM10-preferring metalloprotease inhibitor GI254023X (Hundhausen *et al*, 2003; Ludwig

*et al*, 2005) (Fig 1B) as well as the conditional knock-out of ADAM10 (Fig 1C) abolished secretion of the shed NrCAM ectodomain (sNrCAM), as well as of sAPPα into the conditioned medium of primary, murine neurons and increased full-length, mature NrCAM (mNrCAM) levels in the neuronal lysate. The protein bands were specific for sNrCAM and mNrCAM as demonstrated with shRNA knock-down experiments of NrCAM and using different antibodies (Fig EV1A). This demonstrates that NrCAM is a substrate for ADAM10 in neurons and is consistent with previous studies using cultured neurons *in vitro* (Kuhn *et al*, 2016; Brummer *et al*, 2018). In contrast, the conditional knock-out of ADAM17, a close homolog of ADAM10, did not alter NrCAM proteolysis in primary neurons (Fig EV1B–D), demonstrating the specific requirement of ADAM10 for NrCAM shedding. Taken together, we conclude that NrCAM is a substrate for ADAM10, which, unlike other substrates (e.g., APP, L1CAM), is not additionally cleaved by ADAM17.

#### NrCAM is processed to a heterodimer and then cleaved by ADAM10

The ADAM10-cleaved sNrCAM lacks the transmembrane and cytoplasmic domains of full-length, mature mNrCAM (Fig 1A) and thus should have a lower apparent molecular weight on immunoblots than mNrCAM in the lysate. Yet, both had the same apparent molecular weight of 150 kDa (Fig 1A). This likely results from a previously described furin protease cleavage site located within NrCAM's third FNIII domain (scheme in Fig 1A) (Kayyem *et al*, 1992; Davis & Bennett, 1994; Susuki *et al*, 2013) and, thus, further away from the membrane than the membrane-proximal ADAM10 cleavage site. The furin cleavage is expected to occur in the secretory pathway, converting the full-length pro-form of NrCAM (proNrCAM) to the cleaved mature mNrCAM. As a result, mNrCAM may be present as a heterodimer consisting of the large N-terminal ectodomain fragment that ends at the furin site, and a C-terminal fragment (CTFf) that spans from the furin cleavage site to the cytoplasmic C-terminus of NrCAM (Fig 1A). In fact, immunoprecipitation of NrCAM with an N-terminally binding antibody co-precipitated the CTFf fragment (Fig EV2A). Conversely, immunoprecipitation of NrCAM with a C-terminally binding antibody co-precipitated the N-terminal furin-cleaved NrCAM ectodomain (Fig EV2B). Together, these results demonstrate that mNrCAM is a heterodimer of a 150 kDa N-terminal ectodomain fragment and the 60 kDa CTFf fragment. A similar heterodimer formation is known for Notch1, another ADAM10 substrate (Sanchez-Irizarry *et al*, 2004).

The furin inhibitor dec-RVKR-CMK reduced maturation of NrCAM in primary murine neurons, but did not affect shedding (Fig EV2C), ruling out an involvement of furin in NrCAM shedding. Moreover, only the ectodomain of the mature, furin-cleaved NrCAM was detected in the conditioned medium of untreated controls. In agreement with previous publications (Anders *et al*, 2001; Lopez-Perez *et al*, 2001), dec-RVKR-CMK decreased both mADAM10 and sAPPα levels (Fig EV2D). Additionally, it decreased sNrCAM, which results from furin cleavage, and increased instead of the cleavage of the non-furin-cleaved soluble form of NrCAM (s-proNrCAM). Thus, total sNrCAM (sNrCAM + s-proNrCAM) levels remained unaltered (Fig EV2C), in agreement with a previous study (Suzuki *et al*, 2012). This suggests that small changes in mADAM10 levels affect APP shedding more strongly than NrCAM shedding. Together with prior results, using NrCAM mutants carrying the mutated furin cleavage site (Suzuki *et al*, 2012), we conclude that under non-inhibited

**Figure 1.  ADAM10 is required for NrCAM shedding in primary neurons.**

A   Schematic diagram of NrCAM's domain structure and sequential proteolytic processing. NrCAM is firstly cleaved by furin, then by ADAM10, and finally by the γ-secretase. The red antibody indicates the binding region of the N-terminal NrCAM antibody. ICD (intracellular domain), CTFf (C-terminal fragment created by furin cleavage), CTFα (C-terminal fragment created by ADAM10 cleavage).

B   Detection of soluble, cleaved NrCAM (sNrCAM) and full-length, mature NrCAM (mNrCAM) and soluble APPα (sAPPα) in neuronal supernatants and lysates (prepared from E16 neurons), treated with GI254023x (5 μM), or solvent for 48 h.

C   Detection of sAPPα, sNrCAM, and mNrCAM in ADAM10 cKO neuronal supernatants and lysates at 7 days *in vitro* (DIV7). The neurons prepared from floxed ADAM10 (ADAM10fl/fl) mice were infected with a lentivirus encoding improved Cre recombinase (iCre) or a control GFP lentivirus at DIV2. Conditioned media were collected for 48 h.

Data information: In (B and C), densitometric quantifications of the Western blots are shown on the right (**$P < 0.01$; ***$P < 0.001$; ****$P < 0.0001$, two-sided Student's *t*-test, *n* = 6–8). Given are mean ± the standard error of the mean. The mean levels of solvent-treated cells were set to 1. Representative Western blots are shown. Source data are available online for this figure.

conditions, NrCAM first undergoes cleavage by furin and subsequently by ADAM10.

### NrCAM is a γ-secretase substrate

After ADAM10 cleavage, the remaining membrane-bound C-terminal fragments of several type I membrane proteins, such as APP and L1CAM, are further processed by the γ-secretase complex, an intramembrane protease (Lichtenthaler *et al*, 2011). As a result, a short N-terminal peptide—comprising the short remaining ectodomain and half of the transmembrane domain—is typically secreted from cells, whereas the intracellular domain (ICD) is released into the cytosol, where it is mostly rapidly degraded. Inhibition of γ-secretase, for example, with the established inhibitor DAPT (Dovey *et al*, 2001), blocks ICD generation and leads to increased CTF levels (for a schematic, see Fig 1A), which is used as a read-out to monitor whether a membrane protein is a substrate for γ-secretase. In order to examine whether the type I membrane protein NrCAM is also cleaved by γ-secretase, we analyzed HEK293 cells that had been transiently transfected with a C-terminally VSV-tagged human NrCAM construct. As expected, NrCAM maturation and sNrCAM shedding by ADAM10 were not affected by inhibition of γ-secretase with DAPT (Fig 2). In contrast, DAPT led to the appearance of a C-terminal NrCAM fragment (termed CTFα) with a molecular weight of around 27 kDa and was not seen upon additional inhibition of ADAM10 with GI254023X (Fig 2) which is in agreement with the fragment ranging from the ADAM10 cleavage site to the C-terminus of NrCAM. As a control, the fragment was barely seen in control-treated cells and not seen in control-transfected cells not expressing NrCAM. Taken together, these findings demonstrate that NrCAM is not only a substrate for ADAM10, but also for γ-secretase.

### ADAM10 controls surface levels of NrCAM and neurite outgrowth

Next, we tested whether loss of ADAM10 cleavage increased surface NrCAM levels and altered neurite outgrowth *in vitro*, which is one of the established functions of NrCAM as a neuronal surface co/receptor (Morales *et al*, 1993; Sakurai *et al*, 1997; Falk *et al*, 2004; Zelina *et al*, 2005; Nawabi *et al*, 2010; Torre *et al*, 2010; Demyanenko *et al*, 2011, 2014; Kuwajima *et al*, 2012; Dai *et al*, 2013). Indeed, surface mNrCAM levels were increased by about 50% in conditional ADAM10 knock-out neurons compared to control-transduced neurons, as established by cell surface biotinylation (Figs 3A and EV3A). The same result was obtained in wild-type neurons treated with the ADAM10-preferring inhibitor GI254023X (Fig 3B). Importantly, ADAM10 inhibition did not alter mRNA levels of either ADAM10 or NrCAM (Fig EV3D). As a control, levels of the unrelated kainate receptor GluR6/7, which is not a substrate of ADAM10 (Kuhn *et al*, 2016), were used as negative control in the surface biotinylation assay and did not show altered surface levels upon inhibition or knock-out of ADAM10 (Fig 3B). Taken together, these results reveal that ADAM10 controls cell surface abundance of NrCAM in primary cortical neurons.

To test whether this affects neurite outgrowth in an NrCAM-dependent manner, neurite growth of single cortical neurons was traced in microfluidic chambers, where neurite and soma compartment were separated (for an overview of the time line, see Fig 3C). Neurons were infected 4 h after plating with a lentivirus encoding GFP (to visualize the neurons) together with two different shRNA

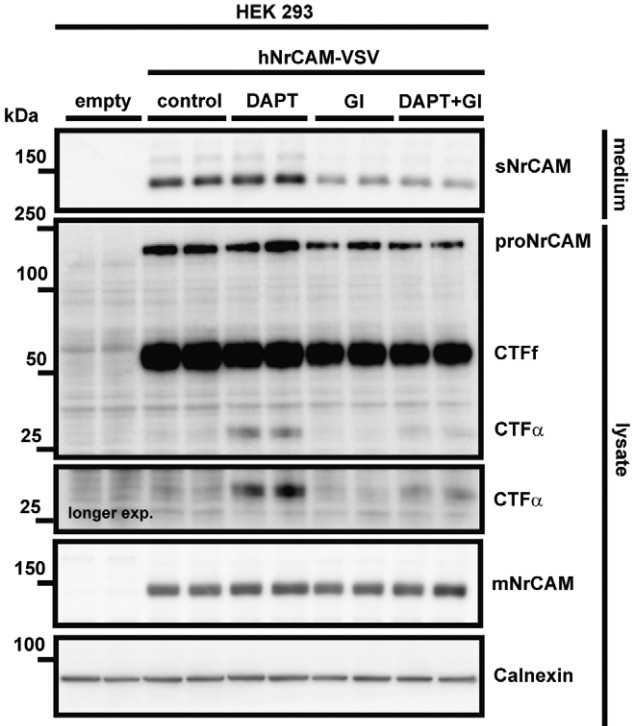

**Figure 2. NrCAM is a γ-secretase substrate.**

HEK 293 cells were transfected with a C-terminally VSV-tagged NrCAM construct or empty vector. Cells were then treated with DAPT (1 μM), GI254023x (5 μM), either substances or solvent for 24 h. The C-terminal NrCAM fragment was detected with an antibody against the VSV-tag. Representative Western blots are shown.

Source data are available online for this figure.

expression cassettes targeting either NrCAM (Fig EV3B and C) or carrying a scrambled control construct. ADAM10 inhibition by GI254023X doubled the length of the neurite outgrowth within 24 h. Knock-down of NrCAM (with shRNAs 1 and 2) decreased neurite outgrowth compared to the control-treated cells by about 50% (Fig 3D and E). Addition of the ADAM10-preferring inhibitor GI254023X was not able to rescue the reduced neurite outgrowth. Taken together, the results reveal that ADAM10 controls neuronal surface levels of NrCAM and neurite outgrowth in an NrCAM-dependent manner.

### Acitretin increases APP but not NrCAM shedding in primary neurons and AD patients

A mild stimulation of APP shedding by ADAM10 is considered as a possible treatment strategy for AD and has been successfully tested in a recent phase II clinical trial, where acitretin indeed mildly increased sAPPα levels by 25% (Endres *et al*, 2014). Effects on other ADAM10 substrates or on cognitive measures of the treated patients were not observed. Major mechanism-based side effects were not seen in that study. However, they remain a concern for future, larger clinical studies given the broad spectrum of neuronal ADAM10 substrates (Kuhn *et al*, 2016), including NrCAM and the role of its cleavage by ADAM10 in neurite outgrowth as

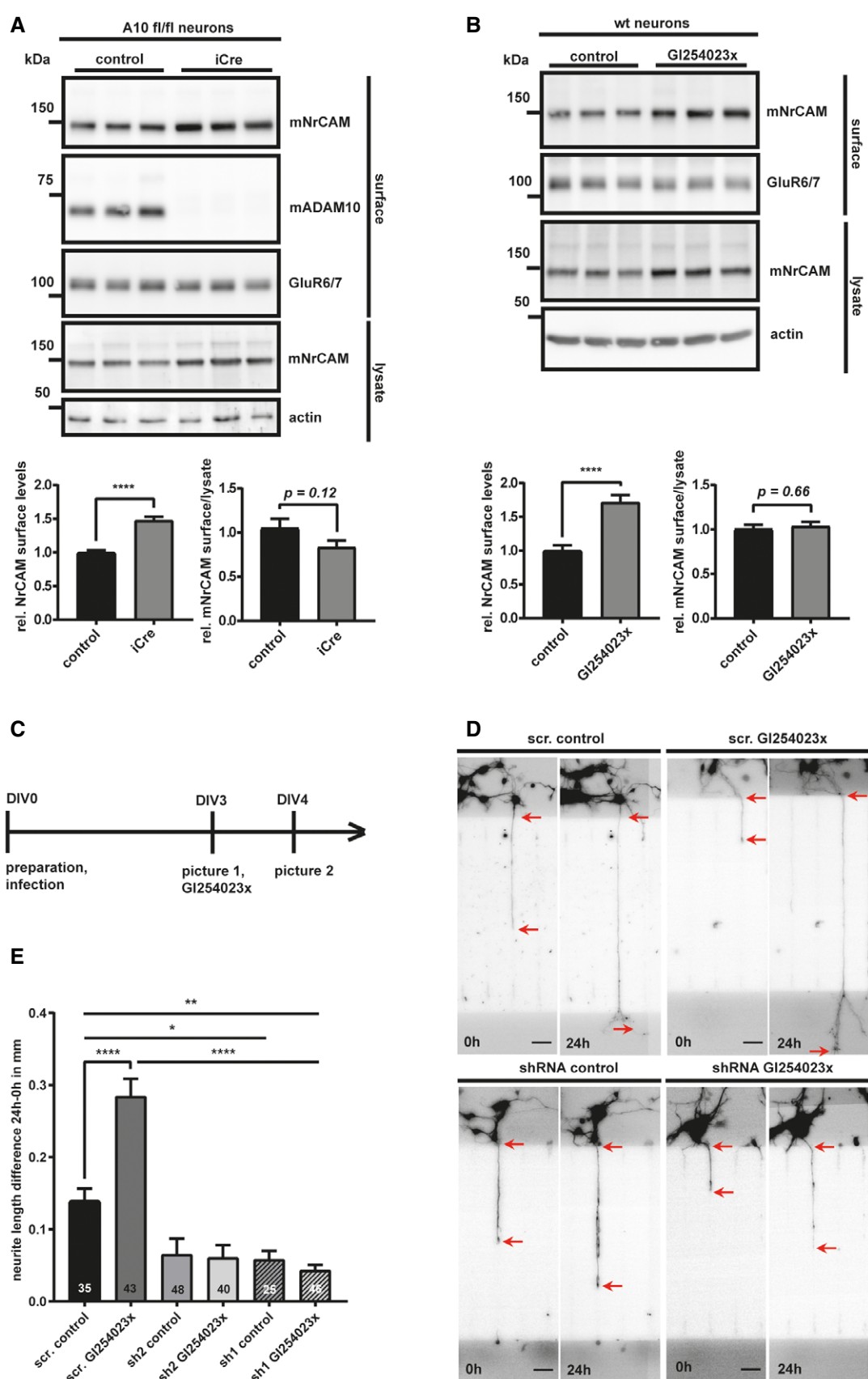

**Figure 3.**

**Figure 3.   ADAM10 is controlling surface expression of NrCAM and NrCAM-mediated neurite outgrowth.**

A   ADAM10fl/fl neurons were treated with an iCre (to induce an ADAM10 KO), or a control lentivirus at DIV2. At DIV7, cell surface proteins were labeled with biotin and enriched by streptavidin pull-down. The biotinylated proteins were detected by immunoblotting. Total lysates were analyzed to compare mNrCAM's surface/lysate levels. Densitometric quantifications of the Western blots are shown. Two-sided Student's t-test (****P < 0.0001, n = 7). Given are mean ± the standard error of the mean. The mean levels of solvent-treated cells were set to 1.

B   Primary murine neurons were treated with GI254023x (5 μM) or solvent, for 48 h. The cell surface proteins were enriched like in (A). Densitometric quantifications of the Western blots are shown. Two-sided Student's t-test (****P < 0.0001, n = 11). Given are mean ± the standard error of the mean. The mean levels of solvent-treated cells were set to 1. Representative Western blots are shown.

C   Workflow of the neurite outgrowth assay.

D   Representative pictures of single neurites at timepoint 0 h (DIV3) and 24 h (DIV4). Neurons were infected 4 h after plating with lentiviruses encoding GFP (to visualize the neurons) together with shRNA expression cassettes (sh1 and sh2) targeting either NrCAM or carrying a scrambled (scr) control construct. Images of neurites were taken at 3 days in vitro (DIV3) and 24 h later at DIV4. In order to study the effect of ADAM10 on neurite outgrowth, neurons were treated with the ADAM10 inhibitor GI254023x, or vehicle (control), at DIV3, after taking the first pictures with an epifluorescent microscope. The differences in neurite length were calculated as absolute values (neurite length at 24 h minus neurite length of 0 h) for individual neurites passing through the middle channels of the chambers. Only neurites that had already entered the main channel at 0 h and had not yet left those channels at 0 h were considered. The red arrows indicate the start and the end of the respective length measurements. The scale bar indicates 40 μm.

E   Quantification and statistical analysis of the neurite outgrowth assay shown in (D). Scr. = scrambled; sh 1 and 2 = shRNA1 and 2. One-way ANOVA with post hoc Dunnett's test. Given are mean ± the standard error of the mean (*P < 0.05; **P < 0.01; ****P < 0.0001, n-numbers for each condition are shown in the graph).

Source data are available online for this figure.

demonstrated above. Thus, sNrCAM may potentially serve as a biomarker for drug responses in acitretin trials. While our experiments so far demonstrated that sNrCAM is a suitable substrate marker to monitor decreases in ADAM10 activity, we now tested whether NrCAM may also be used to monitor increases in ADAM10 activity and, thus, may potentially serve as a biomarker for drug responses in acitretin trials. To this aim, we treated primary neurons with or without acitretin. As a control, we first monitored levels of sAPPα. As expected and in agreement with previous studies (Tippmann et al, 2009; Endres et al, 2014), acitretin increased sAPPα levels in the conditioned medium of murine and rat neurons and ADAM10 protein levels in the neuronal lysate (Figs 4A and B, and EV4A and B). ADAM10 was also increased at the mRNA level (Fig EV4A), as determined in the murine neurons and in line with previous studies (Tippmann et al, 2009; Endres et al, 2014). Surprisingly, however, acitretin did not increase sNrCAM levels (Fig 4A and B). Moreover, NrCAM mRNA levels, as well as full-length NrCAM protein (proNrCAM and mNrCAM), were also not affected by acitretin treatment (Figs 4A and B, and EV4A). Moreover, shedding of sMT4-MMP, another ADAM10 substrate (Kuhn et al, 2016), was also not altered upon acitretin treatment, suggesting a relatively specific enhancing effect on APP shedding. To corroborate these findings in vivo, we analyzed human CSF samples from the recent acitretin phase II clinical study. sNrCAM levels were compared by immunoblots in individual patients before and after treatment (Fig 4C and D). Albumin, which is endogenously present in human CSF, was used as a loading control. Similar to the primary neuron experiment in vitro, acitretin treatment did not increase sNrCAM levels in the patients' CSF (Fig 4D), while sAPPα was increased by 1.25-fold in the same samples, as reported previously (Endres et al, 2014). This unexpected finding demonstrates that it is possible to activate the ADAM10-mediated cleavage of APP without interfering with the shedding of NrCAM, another ADAM10 substrate.

## NrCAM shedding can be stimulated with other ADAM10 cleavage activators

To rule out the possibility that acitretin did not increase NrCAM shedding, simply because NrCAM shedding is already at the

maximum and cannot be further increased, we did a further control experiment to demonstrate that NrCAM shedding can be activated with a suitable stimulus. We chose NMDA, which is known to increase ADAM10's synaptic localization and activity (Maretzky et al, 2005; Marcello et al, 2007; Wan et al, 2012) and consequently ADAM10 activity toward synaptic substrates, such as APP, Nectin-1, or Neuroligin-1 (Marcello et al, 2007; Kim et al, 2010; Suzuki et al, 2012). Primary cortical neurons were treated with NMDA, GI254023X, or both for 30 min, which resulted in a strong increase in sNrCAM levels and a mild concomitant decrease of mNrCAM in the lysate (Fig 5A). Importantly, this effect was prevented when the neurons were additionally treated with GI254023X, demonstrating that NMDA activated the cleavage of NrCAM through ADAM10 (Fig 5A). The same result was obtained using ADAM10 knock-out neurons (Fig EV5A). As a control, the NMDA-induced increase in NrCAM cleavage was blocked with the specific NMDA receptor antagonist D-APV (Fig EV5B).

Finally, we asked whether the NMDA-activated ADAM10 cleavage of NrCAM would control surface levels of NrCAM in a manner similar to Fig 3, where we found that also constitutive ADAM10 cleavage is a mechanism to regulate NrCAM surface levels. To this aim, we performed cell surface biotinylation of cortical neurons, stimulated with NMDA for 30 min. Treatment with NMDA significantly decreased the surface expression of mNrCAM by nearly 75% (Fig 5B), while only mildly reducing mNrCAM levels in the total lysate. This suggests that the short-term treatment with NMDA predominantly induces sNrCAM shedding at the cell surface.

We conclude that ADAM10 is not only the constitutive, but also a stimulated NrCAM sheddase and that ADAM10 controls cell surface levels of NrCAM both under basal, non-stimulated conditions and upon neuronal stimulation with NMDA.

## Acitretin selectively releases substrates and non-substrates into the CSF of Alzheimer patients

The experiments above revealed that acitretin selectively activated ADAM10 cleavage of APP, but not of NrCAM. If acitretin would also spare other ADAM10 substrates, this would be a major

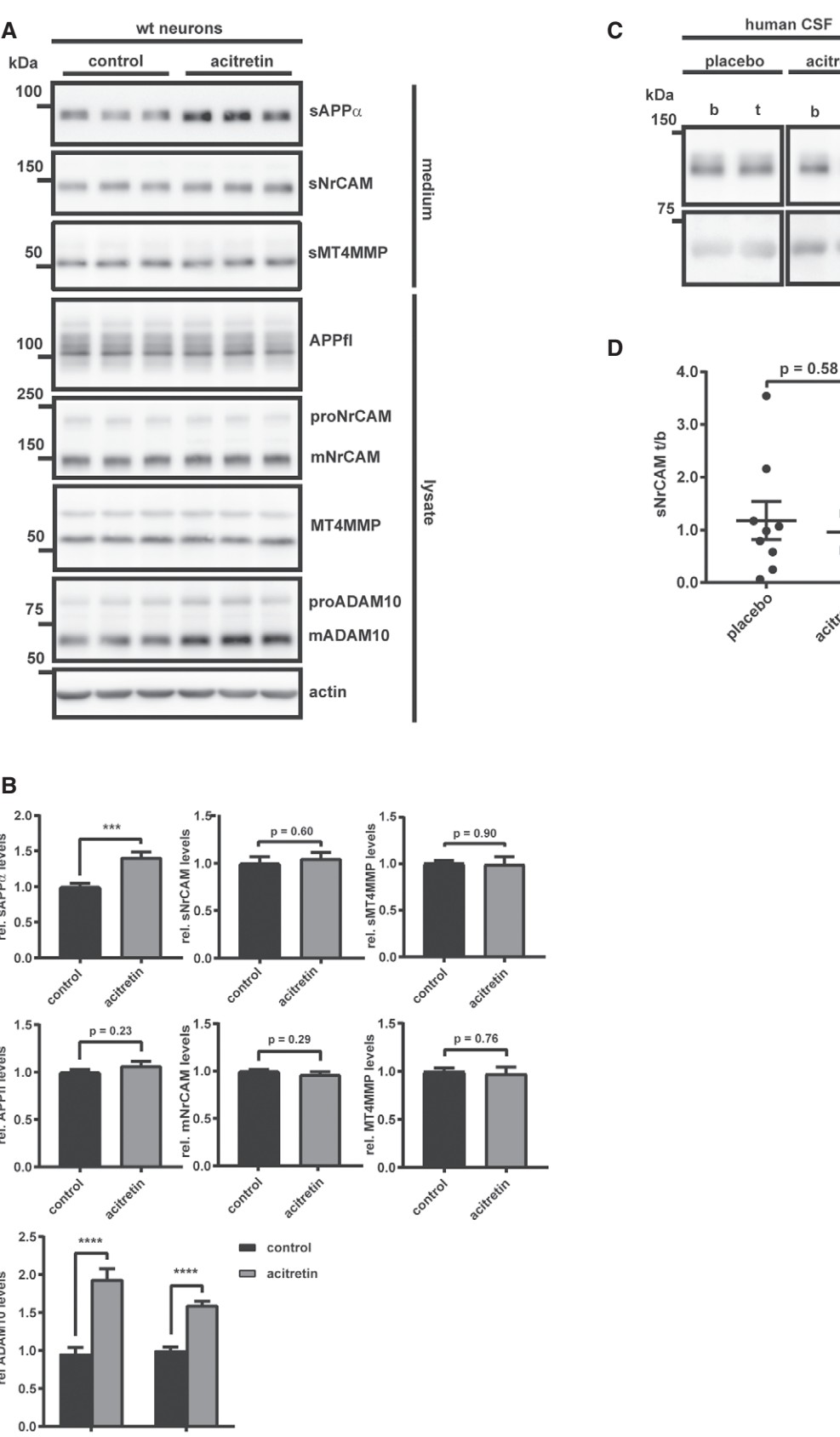

**Figure 4.**

**Figure 4.  Acitretin increases APP but not NrCAM shedding in primary neurons and AD patients.**

A    Wt neurons were kept in culture until DIV5; then, the cells were treated with acitretin (4 μM) or vehicle control for 48 h, as described earlier (Tippmann et al, 2009).

B    Densitometric quantifications of the Western blots are shown (***P < 0.001; ****P < 0. 0001, two-sided Student's t-test, n = 9–20). The mean levels of solvent-treated cells were set to 1.

C    Detection of sNrCAM and human serum albumin (hSA) in AD patients CSF that had been treated with acitretin (30 mg/day) or placebo for 30 days (n = 9) (Endres et al, 2014). b = baseline; t = treatment.

D    Densitometric quantification of the Western blots in (C). sNrCAM/hSA ratios were calculated, and then, the treatment-to-baseline (t/b) ratios for every patient were calculated. Two-sided Student's t-test (n = 9).

Data information: Given are mean ± the standard error of the mean. Representative Western blots are shown.

Source data are available online for this figure.

advantage for AD treatment as it may reduce potential side effects resulting from increased cleavage of other known ADAM10 substrates. Thus, we systematically tested the effect of acitretin on the CSF proteome of AD patients. We obtained 18 CSF samples from the recent clinical phase II trial (Endres et al, 2014) that were analyzed by immunoblot for NrCAM shedding in Fig 4C. Protein intensities after treatment (t) with either acitretin or placebo were divided by the corresponding baseline (b) protein intensities before treatment (t/b ratio) for every patient. To determine acitretin-dependent effects, the baseline normalized protein intensities (t/b ratios) of the acitretin-treated group were compared to the placebo group. The results are displayed in a volcano plot in which the minus log10-transformed P-value is plotted against the log2-transformed average ratio of the acitretin and placebo group (Fig 6). Given the overall mild changes, we did not apply multiple hypothesis corrections, for example, using a Bonferroni test. Instead, proteins were considered as potential hits if their significance P-value was < 0.05 using a Student's t-test and had levels that were up- or down-regulated in the acitretin group by more than 20% compared to the controls and thus, in a similar range as sAPPα, which was reported to be increased in the same samples by 25% (Endres et al, 2014). This yielded 8 proteins (SERPINA7, PTPRS, CFHR1, B4GALT1, ST6GAL2, BCAM, GDA, and KLK11) being down-regulated and 6 proteins (DCD, FABP5, PDCD6, TMSB4X, CSPG4, and CST6) being up-regulated (Fig 6, Table 1). Surprisingly, only one previously known ADAM10 substrate, CSPG4/NG2, which is expressed in oligodendrocyte precursor cells (Sakry et al, 2014), was found among the proteins with increased levels (Fig 6), while over 50 known ADAM10 substrates (Dataset EV1) did not show significant changes, including NrCAM, thus confirming the Western blot results from Fig 4C and D. Three additional ADAM10 substrates, ST6GAL2 (Kuhn et al, 2016), BCAM (Cai et al, 2016), and PTPRS (Kuhn et al, 2016), even showed a lower abundance than in the placebo group (see also Table 1), clearly showing that acitretin did not increase the shedding of all ADAM10 substrates. Additionally, it is important to note that the extent of the changes was mild for the one substrate with increased (22%) and the three substrates with decreased (up to 40%) ADAM10 cleavage products. While acitretin increased sAPPα in the CSF (Endres et al, 2014) and in our neuronal experiments (Fig 4A), total levels of sAPP were not altered in the CSF, so that APP was not detected as a hit in the proteomic analysis (Fig 6). This result was expected because total sAPP does not only comprise sAPPα, but also all other sAPP forms generated by other proteases, including BACE1 and δ-secretase. These individual sAPP species can well be detected by immunoblots using cleavage site-specific antibodies, but not by mass spectrometry, where all tryptic peptides of APP

are used for quantification. We consider the possibility that some of the additional ADAM10 substrates may have—similar to APP—mildly increased ADAM10 cleavage and a compensating reduction in cleavage by other proteases, such that total cleavage was not altered with acitretin in the proteomic study.

Besides the few ADAM10 substrates, 10 proteins had increased (DCD, FABP5, PDCD6, TMSB4X, and CST6) or reduced (SERPINA7, CFHR1, B4GALT1, GDA, and KLK11) CSF levels. Because they are not membrane proteins with the exception of B4GALT1, they are unlikely to be direct ADAM10 substrates. Instead, they may be changed as an indirect effect of ADAM10 activation or of the drug acitretin itself. Interestingly, B4GALT1 is a known substrate for the protease SPPL3 (Kuhn et al, 2015), so that the reduced CSF levels also appear as an indirect effect of the acitretin treatment. Taken together, the proteomic analysis of the clinical CSF samples reveals that acitretin only mildly affected the CSF protein composition of AD patients and demonstrates that the acitretin-induced ADAM10 activation is presumably moderate in vivo, as only few ADAM10 substrates showed altered protein levels in CSF.

### TSPAN15 has opposite effects on ADAM10-mediated shedding of NrCAM and APP

We next wished to provide a better understanding of the unexpected finding that the general ADAM10 activator acitretin only stimulates shedding of APP and CSPG4/NG2 in the patient CSF, but not of NrCAM and many other ADAM10 substrates. ADAM10 binds to six different membrane proteins of the tetraspanin (TSPAN) family and it has been suggested that they control the substrate specificity of ADAM10. To test whether a tetraspanin may indeed have differential effects on APP and NrCAM shedding, we used TSPAN15 (Matthews et al, 2017), which is the best studied family member among the ADAM10 interactors (Haining et al, 2012; Jouannet et al, 2016; Noy et al, 2016; Seipold et al, 2018). TSPAN15 or empty control vector was co-transfected with NrCAM into HEK293 cells, which only express low levels of endogenous TSPAN15. Similar to acitretin, TSPAN15 increased mADAM10 levels by about 50% (Fig 7), which is in line with previous reports (Haining et al, 2012; Noy et al, 2016). Interestingly, TSPAN15 significantly increased the shedding of sNrCAM (twofold), while it decreased the release of sAPPα (threefold), showing that TSPAN15 does indeed differentially regulate NrCAM and APP cleavage. Furthermore, these results show that an increase in mADAM10 does not necessarily increase the shedding of all of its substrates.

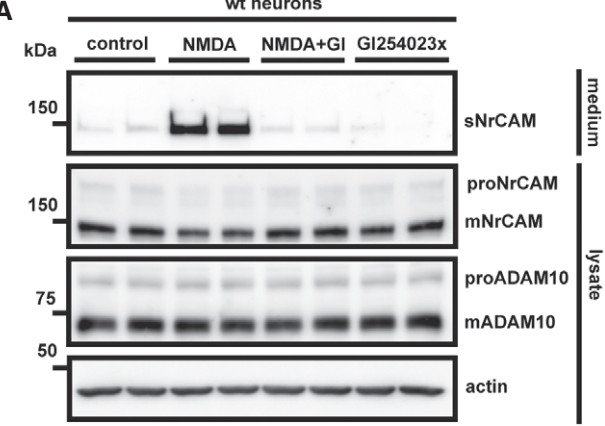

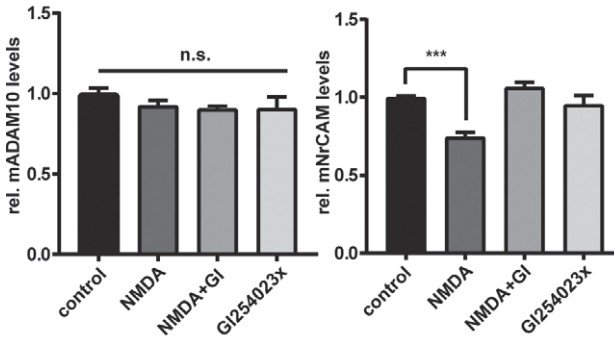

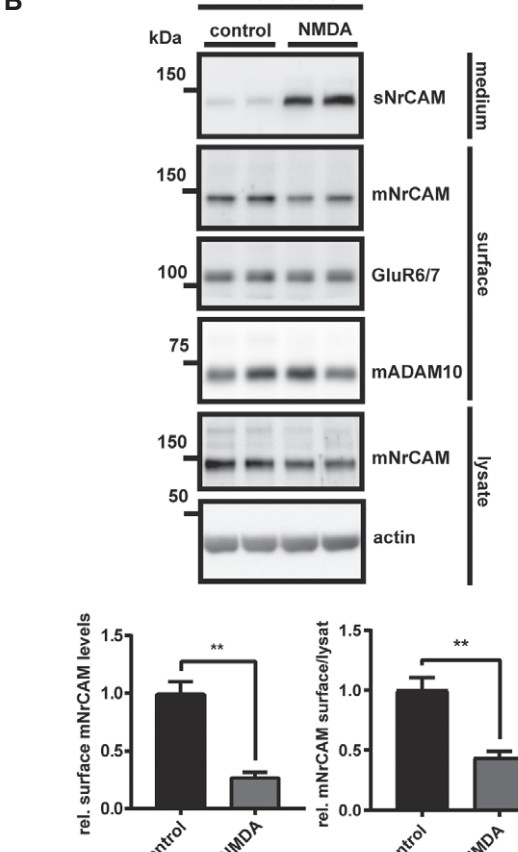

**Figure 5.    Neuronal stimulation by NMDA increases ADAM10-mediated NrCAM shedding.**

A    To ensure proper synapse formation, primary neurons were kept in culture for 10 days prior to the treatment (Wan *et al*, 2012). Then, cells were treated with NMDA (50 μM), NMDA (50 μM) and GI254023x (5 μM), GI254023x (5 μM) alone or vehicle for 30 min. Total cellular mADAM10 levels remained unchanged by the treatment, which is in line with NMDA altering mainly the intracellular localization of ADAM10 by driving the protein to the synaptic membranes (Marcello *et al*, 2007). Densitometric quantifications of the Western blots are shown. One-way ANOVA with *post hoc* Dunnett's test (***$P < 0.001$, $n = 7$). Given are mean ± the standard error of the mean. The mean levels of solvent-treated cells were set to 1. The membrane was reprobed with the different indicated antibodies.

B    Wt neurons were kept in culture until DIV10; then, the cells were treated with NMDA (50 μM), or vehicle for 30 min. Cell surface proteins were labeled with biotin and enriched by streptavidin pull-down. The biotinylated proteins were detected by immunoblotting. Total lysates were analyzed to compare mNrCAMs surface/lysate levels. Densitometric quantifications of the Western blots are shown. Two-sided Student's *t*-test (**$P < 0.01$, $n = 6$). Given are mean ± the standard error of the mean. The mean levels of solvent-treated cells were set to 1. Representative Western blots are shown.

Source data are available online for this figure.

## Discussion

Our study establishes the cell adhesion protein NrCAM as a physiological substrate for the α-secretase ADAM10 and reveals a role for ADAM10 in neurite outgrowth through NrCAM cleavage. Furthermore, our study demonstrates that soluble, ADAM10-cleaved sNrCAM is an excellent marker in AD patients for developing and testing drugs that selectively activate cleavage of APP over other ADAM10 substrates. Finally, whole proteome analysis of patient samples revealed that acitretin in a clinical trial surprisingly activates ADAM10 in a substrate-selective manner and, thus, may be safer in patients than expected.

ADAM10 is a drug target in AD, where an activation of the ADAM10 cleavage of APP is therapeutically desired in order to reduce generation of the neurotoxic Aβ peptide (Postina, 2012; Marcello *et al*, 2017). Importantly, a mild increase of only 30% of mature ADAM10 levels in mouse brains was sufficient to lower amyloid β levels and prevented amyloid pathology in an AD mouse model (Postina *et al*, 2004). Yet, ADAM10 has numerous additional substrates in the brain and in other organs (Crawford *et al*, 2009; Saftig & Lichtenthaler, 2015; Kuhn *et al*, 2016), which may cause mechanism-based side effects upon therapeutic ADAM10 activation. The second generation retinoid acitretin, which is in clinical use to treat psoriasis, mildly increases ADAM10 levels *in vitro* by 1.3-fold (Tippmann *et al*, 2009) and sAPPα levels in human CSF by 25% (Endres *et al*, 2014) and, thus, was expected to also increase CSF levels of the cleaved ectodomain of other ADAM10 substrates in the brain. Surprisingly, however, this was not the case. Only one previously known ADAM10 substrate (CSPG4/NG2) (Sakry *et al*, 2014) had mildly increased CSF levels (22%)—similar to sAPPα—, while more than 50 previously identified ADAM10 substrates or substrate candidates, including sNrCAM, did not show significantly altered levels in CSF. Interestingly, three known ADAM10 substrates (BCAM, ST6Gal2, and PTPRS) even had slightly reduced CSF levels. CSPG4/NG2 (Sakry *et al*, 2014), which was the only up-regulated substrate apart from sAPPα (Endres *et al*, 2014), is mainly

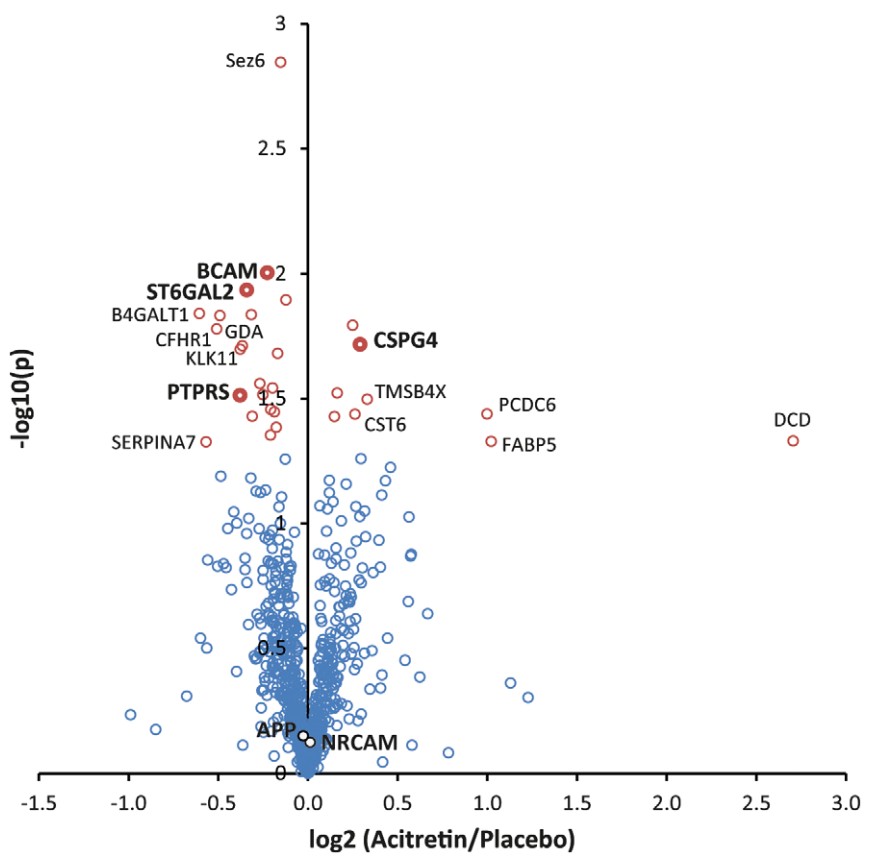

**Figure 6. Mass spectrometric analysis of acitretin CSF samples.**

Volcano plot of the proteomic analysis of CSF from AD patients treated with acitretin or placebo for 30 days. Every circle represents one protein. The log10-transformed *t*-test *P*-values (again treatment/baseline for every patient) are plotted against the log2-transformed label-free quantification intensity ratios of acitretin and placebo CSF. Proteins with a *P*-value of less than 0.05 are labeled in red. Proteins that are additionally changed by more than 20% compared to placebo are labeled with their names. ADAM10 substrates are indicated in bold writing. In total, more than 50 ADAM10 substrates were identified, but only the significantly altered substrates as well as NrCAM and APP are indicated.

expressed in oligodendrocyte precursor cells. CSPG4 is an important regulator for adaptive processes following brain damage (Schafer & Tegeder, 2018) and is able to promote neurite regrowth into glial scar tissue (Vadivelu *et al*, 2015).

ADAM10 cleaves membrane proteins, but several of the affected proteins were not membrane proteins and, thus, their levels are unlikely to be changed because of an increased, direct cleavage by ADAM10. Yet, their changes may be secondary consequences of ADAM10 activation, as ADAM10 activity is required for different signaling pathways, including Notch signaling (Pan & Rubin, 1997). This assumption is in agreement with a previous study demonstrating that mild (30%) neuronal overexpression of ADAM10 in mouse brain altered expression of up to 355 genes, although their RNA levels were mostly only moderately up- or down-regulated by up to 40% (Prinzen *et al*, 2009). Another mechanism for changes in protein expression may be ADAM10-independent effects of acitretin. For example, FABP5, which had increased CSF levels upon acitretin treatment, is an all-trans retinoic acid (atRA) transport protein (Hohoff *et al*, 1999; Smathers & Petersen, 2011). Its increased levels are likely a direct consequence of acitretin's function in indirectly elevating cellular atRA levels (Tippmann *et al*, 2009), because retinoic acid is known to increase FABP5 expression (Yu *et al*, 2012). The strongest changes were obtained for PDCD6 and DCD. PDCD6

participates in T-cell receptor-, Fas-, and glucocorticoid-induced programmed cell death, while DCD is an antimicrobial protein that is mainly expressed in sweat glands, whereas its N-terminal fragment promotes neural cell survival under conditions of oxidative stress (Jung *et al*, 2001; Tarabykina *et al*, 2004; Dash-Wagh *et al*, 2011; Burian & Schittek, 2015). Whether the function of both proteins is altered in acitretin-treated patients remains to be investigated. Potentially, DCD could be used to monitor increased ADAM10 activity in patients by simply testing their sweat.

The selective CSF increase of the cleavage products of only two ADAM10 substrates (sAPPα, sCSPG4/sNG2) upon acitretin treatment in patients is unexpected, but our study offers insights into the possibly underlying molecular mechanisms. While NrCAM (this study) and APP (Jorissen *et al*, 2010; Kuhn *et al*, 2010) both require ADAM10 for their cleavage under constitutive, non-stimulated conditions, our study demonstrates that activation of NrCAM and APP shedding beyond the constitutive level can occur through different molecular mechanisms. For example, PMA stimulated shedding of APP, but not NrCAM in primary murine neurons (Fig EV1C and D). Conversely, overexpression of the tetraspanin TSPAN15 increased NrCAM shedding, but even reduced APP shedding in HEK293 cells. This finding is particularly remarkable, as expression of TSPAN15, a known ADAM10 binding partner (Haining *et al*,

**Table 1.  Significantly changed proteins in AD patients treated with acitretin.**

| Protein name | Gene name | (Aci./Pla.) | P-value | Reference ADAM10 substrate |
|---|---|---|---|---|
| Decrease | | | | |
| **Basal cell adhesion molecule** | **BCAM** | 0.79 | **0.0116** | **Cai et al (2016)** |
| Beta-1,4-galactosyltransferase 1 | B4GALT1 | 0.66 | 0.0144 | |
| **Beta-galactoside alpha-2,6-sialyltransferase 2** | **ST6GAL2** | 0.71 | **0.0147** | **Kuhn et al (2016)** |
| Complement factor H-related protein 1 | CFHR1 | 0.70 | 0.0166 | |
| Guanine deaminase | GDA | 0.78 | 0.0194 | |
| Kallikrein-11 | KLK11 | 0.77 | 0.0200 | |
| **Receptor-type tyrosine-protein phosphatase S** | **PTPRS** | 0.77 | **0.0306** | **Kuhn et al (2016)** |
| Thyroxine-binding  globulin | SERPINA7 | 0.67 | 0.0470 | |
| Increase | | | | |
| **Chondroitin sulfate proteoglycan 4** | **CSPG4** | 1.22 | **0.0191** | **Sakry et al (2014)** |
| Thymosin beta-4 | TMSB4X | 1.26 | 0.0318 | |
| Programmed cell death protein 6 | PDCD6 | 2.00 | 0.0364 | |
| Cystatin-M | CST6 | 1.20 | 0.0365 | |
| Dermcidin; Survival-promoting peptide | DCD | 6.52 | 0.0467 | |
| Fatty acid-binding protein, epidermal | FABP5 | 2.03 | 0.0469 | |

This table contains all proteins with a *t*-test significance *P*-value of < 0.05 (not corrected for multiple testing) and a change of the protein level of more than 20% compared to placebo-treated controls. ADAM10 substrates are marked in bold.

2012; Prox *et al*, 2012; Noy *et al*, 2016), increased the levels of mature ADAM10, similar to what is known for acitretin treatment (Tippmann *et al*, 2009). Thus, an increase in mature ADAM10 cannot be taken as evidence for increased ADAM10 activity toward all of its substrates. In fact, it has been suggested that the six members of the C8 subgroup of the tetraspanin family of four-pass transmembrane proteins, including TSPAN15, do not only bind to ADAM10, but also to its substrates and thus control the substrate specificity of ADAM10 (Matthews *et al*, 2017). As a result, ADAM10 may be present in six different complexes, with potentially each of them enabling cleavage of different sets of ADAM10 substrates (Jouannet *et al*, 2016; Matthews *et al*, 2017). This assumption is consistent with our finding that TSPAN15 expression had opposite effects on the shedding of NrCAM and APP. Likewise, a recent study demonstrated that TSPAN15 deficiency in mice reduced ADAM10 maturation and selectively reduced ADAM10 cleavage of N-cadherin and the prion protein, whereas APP cleavage by ADAM10 was not affected (Seipold *et al*, 2018). Whether TSPAN15 shows an altered expression in the AD patients that were treated with acitretin is not known, as the brains of these individuals are not available. Yet, while our TSPAN15 experiment served to demonstrate that certain TSPANs can indeed alter ADAM10 cleavage of some substrates versus others, it is unlikely that the acitretin-stimulated, selective increase in sAPPα in the AD patients is directly related to altered TSPAN15 expression, because acitretin (increased sAPPα, no effect on sNrCAM) and TSPAN15 expression (reduced sAPPα and increased sNrCAM) had opposite effects on cleavage of APP and NrCAM. However, it is well possible that acitretin mediates its relatively specific effect through other members of the TSPAN C8 family, which comprises besides TSPAN15 also TSPANs 5, 10, 14, 17, and 33. Among them, especially TSPAN5 and TSPAN14 are highly expressed in particular cell types within the central nervous

system (Matthews *et al*, 2017) and may therefore contribute to the differential acitretin effects on various ADAM10 substrates. In fact, by increasing intracellular atRA levels, acitretin can have an impact on the expression of many genes, not only ADAM10 (Lane & Bailey, 2005). However, generation of mice deficient in or transgenically expressing those TSPANs will be needed to better understand the role of the TSPANs in controlling the substrate selectivity of ADAM10. At this point, it also appears possible that acitretin mediates its effects through proteins other than TSPANs. One precedent comes from the ADAM10-homolog ADAM17, which does not bind TSPANs, but another multi-pass transmembrane protein, iRhom1 or iRhom2. Recent studies revealed that this binary interaction is in fact a ternary interaction with the soluble protein FRMD8/iTAP (Kunzel *et al*, 2018; Oikonomidi *et al*, 2018), which controls stability and activity of the ADAM17/iRhom complex. Thus, it appears possible that additional proteins may also affect ADAM10 binding to TSPANs and thus contribute to substrate selectivity. Such proteins may be alternative targets for acitretin.

Taken together, our current study and the previous studies demonstrate that a substrate-selective activation of ADAM10 is in principle feasible and suggests that additional ADAM10 activators beyond acitretin may be identified in drug development for a future substrate-selective ADAM10 activation in AD. Furthermore, our study provides a possible explanation for the lack of severe side effects upon mild ADAM10 activation.

Another major outcome of our study is the identification of sNrCAM as an excellent marker for measuring substrate-selective ADAM10 activation. For clinical trials in AD, the activation of ADAM10 is measured through increased sAPPα levels and reduced Aβ levels. However, so far it has remained unclear whether other ADAM10 substrates would also be affected. Ideally, the effect of an ADAM10 activator would be measured on all known neuronal

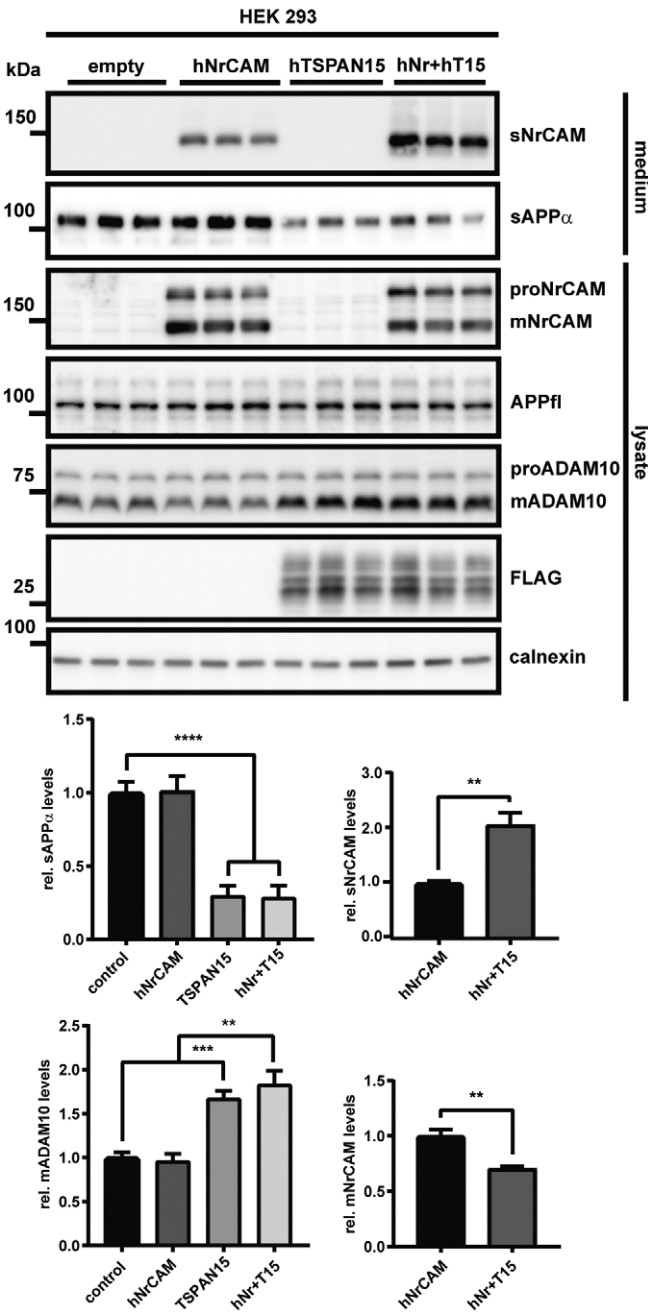

**Figure 7. TSPAN15 has opposite effects on ADAM10-mediated NrCAM and APP shedding.**

HEK293 cells were transiently transfected with hNrCAM, FLAG-tagged hTSPAN15, either plasmids or the empty vector. Detection of sAPPα and sNrCAM in the conditioned media and ADAM10 in the cell lysate. Densitometric quantifications of the Western blots are shown (**$P < 0.01$; ***$P < 0.001$; ****$P < 0.0001$; two-sided Student's $t$-test was used for sNrCAM and mNrCAM and one-way ANOVA with *post hoc* Dunnett's test for sAPPα and mADAM10, $n = 6$). Given are mean ± the standard error of the mean. The mean levels of solvent-treated cells were set to 1. Representative Western blots are shown.

Source data are available online for this figure.

ADAM10 substrates, which may be difficult in a clinical routine setting. Thus, sNrCAM may serve as a proxy for other ADAM10 substrates besides APP, as its cleavage happens mechanistically

differently from APP. Our study demonstrates that, identical to APP, NrCAM requires ADAM10 for its constitutive cleavage, but that an increase of substrate cleavage occurs through different mechanisms for both substrates, APP and NrCAM. Thus, sNrCAM has the potential to be a companion diagnostic for clinical trials with ADAM10 activators, as it allows distinguishing between the intended specific increase in sAPPα and a potential non-desired increase of other ADAM10 substrate cleavage products, including NrCAM. Similar considerations hold true for the two unrelated proteases β- and γ-secretase, which convert APP to the Aβ peptide. For example, γ-secretase inhibitors, which have been discontinued for AD drug development, also block cleavage of numerous additional γ-secretase substrates besides APP, including Notch (Golde *et al*, 2013). Thus, Notch cleavage is used to counter-screen γ-secretase inhibitors to identify such compounds that spare Notch cleavage but still block APP cleavage (Golde *et al*, 2013). Similarly, sNrCAM may be used to identify additional drugs besides acitretin that preferentially activate ADAM10 cleavage of APP.

Our study demonstrates that acitretin did not alter NrCAM shedding *in vitro* or *in vivo*, making it unlikely that the acitretin-mediated increase in ADAM10 leads to mechanism-based side effects resulting specifically from altered NrCAM function. However, our study clearly demonstrates that other ways of modulating ADAM10 levels or specifically altering ADAM10 cleavage of NrCAM have major consequences for neurite outgrowth. While ADAM10 inhibition enhanced full-length NrCAM levels at the neuronal surface and increased neurite outgrowth in the *in vitro* assay, a knock-down of NrCAM abolished the increased neurite outgrowth. This is consistent with previous antibody perturbation experiments which also inhibited neurite outgrowth *in vitro* and additionally disturbed axonal guidance, by interfering with the interaction between NrCAM and its respective ligands at the neuronal or glial surface (Morales *et al*, 1993; Stoeckli & Landmesser, 1995; Sakurai *et al*, 1997; Stoeckli *et al*, 1997). ADAM10 has been shown to differentially regulate neurite outgrowth and axonal guidance through the cleavage of several cell adhesion molecules other than NrCAM (e.g., L1CAM, IgLONs) in distinct anatomical locations and developmental stages (Maretzky *et al*, 2005; Saftig & Lichtenthaler, 2015; Sanz *et al*, 2017). For example, while ADAM10 inhibition decreased neurite growth in cultured DRG neurons/explants and chick retinal ganglion cells, it increased the number of neuritic processes in neuronal SH-SY5Y cells (Paudel *et al*, 2013; Meyer *et al*, 2015; Martins *et al*, 2017; Sanz *et al*, 2017). Likewise, inhibition of L1 shedding in cerebellar microexplant cultures only affected neurite outgrowth, when the cultures were placed on L1-coated coverslips (Maretzky *et al*, 2005). Neurite outgrowth and guidance are complexly regulated and largely rely on the temporal and spatial expression of specific receptors and ligands. Thus, it is not surprising that ADAM10 can have different effects on those processes, depending on the local expression of its respective substrates at specific time points (Kiryushko *et al*, 2004; Maness & Schachner, 2007). This study shows for the first time that ADAM10 can affect neurite outgrowth/length in cortical neurons in an NrCAM-dependent manner. This could be of relevance in neuropsychiatric disorders, such as autism spectrum disorders (ASD) and schizophrenia, to which NrCAM is linked (Marui *et al*, 2009; Ayalew *et al*, 2012) and where neurite outgrowth and branching may be down-regulated by NrCAM shedding in a process impairing neuronal connectivity.

An ADAM10-dependent receptor function for full-length NrCAM is further corroborated by the observation that mice with a conditional knock-out of ADAM10 in neurons (ADAM10 cKO) and NrCAM-deficient mice show opposite phenotypes with regard to dendritic spines. While ADAM10 cKO mice have a reduced number of dendritic spines (Prox *et al*, 2013), NrCAM KO mice have increased dendritic spine densities. Together with Neuropilin-2 and Plexin A3, NrCAM is part of a Sema3F receptor complex, which is mediating a Sema3F-induced spine retraction (Demyanenko *et al*, 2014; Mohan *et al*, 2018). Thus, increased NrCAM surface levels, as we observed upon inhibition of ADAM10, would be expected to decrease spine density, as reported for ADAM10 cKO mice. Conversely, knock-out of NrCAM would increase spine density, as reported (Demyanenko *et al*, 2014). Taken together, these results suggest that alterations of ADAM10 activity, and therefore of NrCAM shedding and surface levels, may be a means of controlling synaptic plasticity, by strengthening the contacts of highly active synapses, while making them less susceptible to Sema3F-induced spine retraction. Yet, given our finding that acitretin did not alter NrCAM shedding, it is likely that the physiological functions of NrCAM are not altered in clinical trials with acitretin.

Besides the function of full-length NrCAM as a surface receptor, as discussed above, the cleavage products—either sNrCAM or the C-terminal fragment of NrCAM or both—may also be functional, because we previously reported an olfactory axon targeting deficit in ADAM10$^{-/-}$ mice, that was similar, if not identical to NrCAM$^{-/-}$ mice (Heyden *et al*, 2008; Kuhn *et al*, 2016). Thus, it appears possible that both full-length NrCAM and its cleavage product(s) may have different functions during development of the nervous system. This is reminiscent of death receptor 6, a member of the TNF receptor superfamily. Full-length DR6 can act as a surface receptor transmitting cell death signals (Nikolaev *et al*, 2009; Mi *et al*, 2011; Strilic *et al*, 2016; Fujikura *et al*, 2017; Gamage *et al*, 2017), while the ADAM10-cleaved DR6 ectodomain can act as a cytokine suppressing proliferation of Schwann cells and delaying myelination in the peripheral nervous system (Colombo *et al*, 2018).

In summary, our study identifies a novel function for ADAM10-dependent processing of NrCAM in regulating neurite outgrowth. Moreover, it establishes the soluble, ADAM10-cleaved sNrCAM as an excellent marker for substrate-selective ADAM10 activation *in vitro* and in patients and reveals that a substrate-selective activation of ADAM10 is feasible in patients. The ability to distinguish between the potentially detrimental activation of ADAM10/NrCAM processing and the protective ADAM10/APP processing will provide new opportunities for safe drug development in AD targeting ADAM10.

# Materials and Methods

## Materials

Antibodies: ADAM10 (1:1,000), ADAM17 (Schlondorff *et al*, 2000) (1:1,000), β-actin (1:1,000), calnexin (1:1,000), N-term. NrCAM (1:1,000), monoclonal human NrCAM (1:500), C-term. NrCAM (1:1000), APP 22C11 (1:1,000), sAPPα 5G11 (1:10) and 14D6 (1:10) (Colombo *et al*, 2013), VSV-G (Santa Cruz) (1:1,000), GluR6/7 (1:1000), hSA (1:1,000), FLAG (1:1,000) (Sigma), and HRP-coupled

anti-mouse and anti-rabbit secondary antibodies (DAKO) (1:10,000). Drugs and reagents: GI254023X (Sigma, 5 μM), acitretin (4 μM), NMDA (50 μM), D-APV (Abcam, 100 μM), dec-CMK (Bachem, 50 μM), PMA (1 μM) recombinant trypsin (Promega), recombinant LysC (Promega), iodoacetamide (Sigma), and dithiothreitol (Biozol).

## Antibodies

| Antibodies | Source; reference |
|---|---|
| Anti-NrCAM (N-terminal) | Abcam (ab24344); (Lustig *et al*, 2001) |
| Anti-NrCAM (C-terminal) | Cell Signaling (55284) |
| Anti-human NrCAM (N-terminal) | R&D (MAB20341); (Wildsmith *et al*, 2014) |
| Anti-APP: 22C11 (N-terminal) | Millipore (MAB384); (Kuhn *et al*, 2010) |
| Anti-sAPPα: 5G11 and 14D6 | (Colombo *et al*, 2013) |
| Anti-ADAM10 (C-terminal) | Abcam (ab124695); (Kuhn *et al*, 2016) |
| Anti-ADAM17 (C-terminal) | (Schlondorff *et al*, 2000) |
| Anti-β-actin | Sigma-Aldrich (A5316); (Pigoni *et al*, 2016) |
| Anti-calnexin | (Enzo, Stressgen, ADI-SPA-860); (Pigoni *et al*, 2016) |
| Anti-flag M2 | Sigma-Aldrich (F1804); (Pigoni *et al*, 2016) |
| Anti-hSA | Sigma-Aldrich (A6684); (Carter-Dawson *et al*, 2010) |
| Anti-GluR 6/7 (GriK 2/3) | Abcam (ab124702); (Kung *et al*, 2013) |
| Anti-VSV-G | Santa Cruz (sc-365019) |

## Plasmids

pcINeo-NrCAM-VSV was kindly provided by Dr. Ben Ze've (Conacci-Sorrell *et al*, 2005). Generation of pEF6A FLAG human TSPAN15 was described previously (Haining *et al*, 2012).

## Mouse strains and cell lines

The following mouse strains were used in this study: wild-type (WT) C57BL/6NCrl (Charles River), ADAM10fl/fl (Prox *et al*, 2013; Kuhn *et al*, 2016), ADAM17fl/fl (Horiuchi *et al*, 2007). All mice were on a C57BL/6 background and were maintained on a 12/12-h light–dark cycle with food and water *ad libitum*. All experimental procedures on animals were performed in accordance with the European Communities Council Directive (2010/63/EU) and in compliance with the German animal welfare law. Embryos were extracted from the pregnant mothers at E15/16. HEK293 cells were purchased from ATCC. HEK293 cells were kept in DMEM supplemented with 10% FBS and 1% penicillin/streptomycin.

## Patients

Patient samples from the prospective, randomized, placebo-controlled phase II trial on acitretin in AD patients were analyzed in this study. The study was initiated after approval by the ethic committees, and executed in accordance with the Good Clinical Practice guidelines (Declaration of Helsinki and International Conference on Harmonization), as described previously (Endres *et al*, 2014). The trial was monitored by the IZKS (University Medical Centre, Mainz) and registered with ClinicalTrials.gov (NCT01078168). Patients provided written informed consent before enrollment.

## Isolation and treatment of primary neurons

Neurons from WT mice were prepared at E15/E16 embryos and cultured as described previously (Mitterreiter *et al*, 2010). The gender of the embryos cannot be determined at this stage. In brief, neurons from wt and ADAM10/ADAM17[Fl/Fl] mice were prepared at E15/E16 and kept in Neurobasal® medium, supplemented with L-glutamine (0.5 mM), 1% penicillin/streptomycin, and B27. At 5 days *in vitro* (DIV), the cells were washed with PBS and the medium was replaced with fresh Neurobasal medium supplemented with L-glutamine (0. 5 mM), 1% penicillin/streptomycin, B27, and the respective drugs. After 48 h of incubation, supernatants were collected and the cells were lysed in STET lysis buffer (50 mM Tris, pH 7.5, 150 mM NaCl, 2 mM EDTA, 1% Triton) that contained GI254023x (5 μM), to prevent an autocatalytic degradation of mADAM10 (Brummer *et al*, 2018). For NMDA treatments, cells were cultured as described earlier (Wan *et al*, 2012). For the treatment with acitretin, the neurons were treated like previously described (Tippmann *et al*, 2009). ADAM10fl/fl and ADAM17fl/fl neurons were infected with iCre, or the control lentivirus at DIV3. For NMDA treatments, cells were cultured as described earlier (Wan *et al*, 2012). For the treatment with acitretin, the neurons were treated like previously described (Tippmann *et al*, 2009). ADAM10fl/fl and ADAM17fl/fl neurons were infected with iCre, or the control lentivirus at DIV3.

## Isolation and treatment of primary rat neurons

All experimental procedures were performed in accordance with the Guidelines for Animal Experiments of the University of Tokyo. Cortical primary neurons from rat were prepared according to a previously described method. In brief, cerebral cortices were dissected from Wistar rat (embryonic day E18), dissociated in Hanks' balanced salt solution (HBSS, Life Technologies, Darmstadt, Germany) including 0.125% trypsin and 5 U/ml DNAse I (both Life Technologies, Darmstadt, Germany) and incubated at 37°C for 15 min. After the addition of DMEM medium containing 10% FBS, 2 mM L-glutamine, and 50 U/ml penicillin/50 μg/ml streptomycin (all Life Technologies, Darmstadt, Germany), cells were pipetted through a cell strainer (100 μm, BD Falcon, Heidelberg, Germany) and subsequently centrifuged at 4°C, 200 *g* for 5 min. Cells were suspended in fresh DMEM culture medium and seeded in a concentration of $1 \times 10^6$ cells/ml on poly-L-ornithine-coated plates (PLO: Sigma, St. Louis, MO, USA). After day 1 *in vitro* (DIV1), medium was exchanged to Neurobasal medium including B27

supplement mix (both Life Technologies, Darmstadt, Germany), 1% glutamine, and 50 U/ml penicillin/50 μg/ml streptomycin. Neurons were cultured for 7 days at 37°C, 5% $CO_2$, and 95% humidity. Cells were treated with acitretin (2 μM) at DIV19, medium was and fresh substances were added every day as described previously (Reinhardt *et al*, 2016).

## Virus production

iCre recombinase lentiviruses were prepared as previously described (Kuhn *et al*, 2016). iCre recombinase lentiviruses were prepared as previously described (Kuhn *et al*, 2016). The following shRNA sequences were used: shRNA1: 5′CGCGTCCGGGGACACCCGTGAGGACTATATCTCGAGATATAGTCCTCACGGGTGTCCTTTTTGGAAA′3; shRNA2: 5′CGCGTCCGGATAGATGGCGATACCATTATACTCGAGTATAATGGTATCGCCATCTATTTTTTGGAAA′3. Targeting sequences, as well as scrambled control sequence, were cloned into plKO2mod-EGFP-WPRE, as previously described (Kuhn *et al*, 2010).

## Transfection

$10^6$ HEK293 cells per well were seeded in a 6-well format and transfected in solution in OptiMEM (Gibco) with the respective vectors (600 ng), using Lipofectamine 2000 (Thermo Fisher)—as to the companies instructions—and incubated for 48 h. Then, the cells were either treated with the respective substances or lysed directly.

## Cell toxicity assay

Wt neurons were placed in a 96-well format (20,000 cells per well); 4 h after plating were either infected with a lentivirus (scr. shRNA-EGFP control, or NrCAM shRNA-EGFP (shRNA1 and shRNA2), 1:1,000) or not. At DIV3, NaOH was added to the cells serving as negative control for cell viability. At DIV4, cell viability was analyzed using the Cell Counting Kit 8 (96992, Sigma-Aldrich) according to the provided instructions. The absorbance was measured with a microplate reader at 450 nm. The individual measurements were performed in 1-h steps, up to 3 h. The values from the initial measurement served as baseline values, and the latter values were normalized on the baseline.

## Cell surface biotinylation

Surface biotinylation was done as described previously (Pigoni *et al*, 2016).

## Cell lysate/supernatant preparation

First, supernatants from HEK293 and neurons were collected. For NMDA (30-min treatment)-, PMA (2-h treatment)-, and acitretin (5 h of conditioned media collection)-treated cells, the supernatants were subjected to TCA precipitation (50 μl TCA/1 ml supernatant) and incubated over-night at 4°C. Then, the samples were centrifuged at 16,000 *g* at 4°C for 5 min, and the pellets were resuspended in 1× Laemmli buffer (2% SDS; 10% glycerol; 0.00625% bromophenol blue; 2.5% β-mercaptoethanol; 31.25 mM Tris; pH 6.8). Cells were washed twice with PBS and lysed in STET lysis buffer (50 mM Tris,

pH 7.5, 150 mM NaCl, 2 mM EDTA, 1% Triton), containing protease inhibitor cocktail (1:500, Sigma, P-8340) and GI254023X (5 μM) (Brummer *et al*, 2018). Samples were centrifuged at 16,000 *g* and 4°C for 5 min, the supernatants were transferred to fresh tubes, and the protein concentration was measured using a BCA (Uptima Interchim, UP95425).

### Quantitative real-time PCRs

NrCAM and ADAM10 qPCR primers were purchased from Bio-Rad (PrimePCR SYBR Green Assay: Adam10, Mouse; PrimePCR SYBR Green Assay: Nrcam, Mouse). Total RNA was extracted using RNeasy Mini Kit (Qiagen) from primary neurons following the manufacturer's instructions. Concentrations and purities of total RNA were spectrophotometrically measured at 260 and 280 nm. Total RNA was reverse transcribed into cDNA, using high-capacity cDNA reverse transcription kit (Applied Biosystems/ABI). Real-time PCR was carried out on a 7500 Fast Real-Time PCR machine (ABI) with the POWER SYBR Green PCR Master Mix (ABI). Reactions were performed in duplicate in 96-well plates. mRNA levels of ADAM10 and NrCAM were normalized on levels of a housekeeping gene (β-actin).

### Co-immunoprecipitations

NrCAM-VSV-transfected HEK293 cells were lysed in CoIP buffer (20 mM HEPES, 150 mM NaCl, 0.5% NP-40, 2 mM EDTA, 10% glycerol) containing protease inhibitor cocktail. Samples were precleared by rotating them with protein G beads (50 μl per 500 μl lysate) at 4°C for 1 h. Then, the samples were pulled down over-night at 4°C with a VSV-G (1:100) or NrCAM antibody (1:100). Samples were centrifuged at 1,500 *g* at 4°C, the supernatant was removed, and the beads were washed twice with CoIP buffer and once with 1× PBS. The samples were then resuspended in 1× Laemmli buffer. The detection was done with the respective opposite antibody.

### Western blotting

The protein concentration was measured using a BCA, and equal protein concentrations for every sample were then subjected to SDS–PAGE separation. Hence, the amount of sample per lane was normalized to the total protein concentration of each sample. 15–20 μg of total protein was used for SDS–PAGE separation. In order to have a second line of control, we also blotted for a housekeeping gene (actin or calnexin), to assure the initial normalization was done properly. Samples were boiled at 95°C for 5 min in reducing (8% SDS; 40% glycerol; 0.025% bromophenol blue; 10% β-mercaptoethanol; 125 mM Tris; pH 6.8) or non-reducing (8% SDS; 40% glycerol; 0.025% bromophenol blue; 125 mM Tris; pH 6.8) Laemmli buffer and then separated on 8% SDS–PAGE gels. The proteins were transferred to PVDF membranes (Millipore) and blocked with 5% milk for 1 h at room temperature. Membranes were incubated for 1 h, or over-night at 4°C with the primary antibody solutions. Then, the membranes were incubated with the secondary antibody for 45 min at room temperature. Membranes were developed with ECL prime (GE Healthcare, RPN2232V1). Western blots were quantified by Multi Gauge software (version 3.0). The values of the band intensity were normalized to the respective control values for each experiment.

### Neurite outgrowth assay

$1 \times 10^5$ neurons were plated into XONA-microfluidic chambers (standard neuron device, SND450) that had been placed on PDL-coated coverslips, according to the manufacturer's instructions. After 4 h, the plated cells were infected with the respective viruses (scr. shRNA-EGFP control, or NrCAM shRNA-EGFP, 1:1000). Cells were kept until DIV3 at 37°C and 5% $CO_2$, and then, the first photomicrographs of the fluorescent neurons were taken with a Leica DM6000 inverted microscope. The images covered the whole channel area in the middle of the respective chambers. Afterward, the neurons were treated with GI254023x (5 μM), or vehicle and kept at 37°C and 5% $CO_2$ for 24 h. At DIV4, a second set of photomicrographs of the same areas were taken and analyzed for length differences of single neurites (length in mm at 24–0 h) with Leica LASX software. Only neurites that had already entered and not yet left the channels on the other side at the timepoint 0 h were used for the calculation. When neurites were separating after leaving the main channel, the longest process was quantified.

### Mass spectrometry

CSF from nine patients per group treated with either acitretin or vehicle control was collected before (baseline value) and after the treatment (Endres *et al*, 2014). The CSF analysis was blinded. A volume of 5 μl per sample was subjected to tryptic digestion followed by liquid chromatography–tandem mass spectrometry (LC-MS/MS) for protein label-free quantification (LFQ) as previously described (Pigoni *et al*, 2016). Nine patients per group were treated with either acitretin or vehicle control. Cerebrospinal fluid was collected before (baseline value) and after treatment. A volume of 5 μl of CSF per sample was subjected to proteolytic digestion in 50 mM ammonium bicarbonate with 0.1% sodium deoxycholate (Sigma-Aldrich, Germany) as previously described (Pigoni *et al*, 2016). Briefly, protein disulfide bonds were reduced with dithiothreitol and sulfhydryl residues were alkylated using iodoacetamide. Proteins were digested using 0.1 μg LysC (Promega) and 0.1 μg trypsin (Promega). Deoxycholate was precipitated by acidification and removed by centrifugation at 16,000 *g* and 4°C for 10 min. Proteolytic peptides were desalted by stop and go extraction (STAGE) with C18 tips (Rappsilber *et al*, 2003). The purified peptides were dried by vacuum centrifugation. Samples were dissolved in 20 μl 0.1% formic acid.

Samples were separated on a nanoLC system (EASY-nLC 1000, Proxeon—part of Thermo Scientific, USA; PRSO-V1 column oven: Sonation, Germany) using an in-house packed C18 column (30 cm × 75 μm ID, ReproSil-Pur 120 C18-AQ, 1.9 μm, Dr. Maisch GmbH, Germany) with a binary gradient of water (A) and acetonitrile (B) containing 0.1% formic acid at 50°C column temperature and a flow of 250 nl/min (0 min, 2% B; 3:30 min, 5% B; 137:30 min, 25% B; 168:30 min, 35% B; 182:30 min, 60% B). The nanoLC was coupled online via a nanospray flex ion source (Proxeon—part of Thermo Scientific, USA) to a Q-Exactive mass spectrometer (Thermo Scientific, USA). Full MS spectra were acquired at a resolution of 70,000. The ten most intense ions exceeding an intensity of $1.5 \times 10^4$ were chosen for collision induced dissociation, and spectra were acquired at a resolution of 17,500. The dynamic exclusion for peptide fragmentation was set to 120 s. The data were

**The paper explained**

**Problem**

The protease ADAM10 is a drug target in Alzheimer's disease (AD), but cleaves numerous substrates in brain, which may cause mechanism-based side effects upon therapeutic ADAM10 activation. This has been a major concern in the past. In addition, the functions of ADAM10 in the brain are still poorly understood, making it hard to predict possible side effects of an ADAM10-activating therapy.

**Results**

We demonstrate a new function for ADAM10 in neuronal development. By cleaving the neuronal cell adhesion protein NrCAM and controlling its cell surface levels, ADAM10 controls neurite outgrowth. Despite this new function, we show that it is unexpectedly possible to selectively increase the ADAM10 cleavage of the Alzheimer APP protein without altering cleavage of all other substrates, including NrCAM. This is shown *in vitro* and also *in vivo* in a clinical trial using the drug acitretin, suggesting that even long-term treatment with acitretin for AD will be safer than expected. Importantly, we demonstrate that the ADAM10-cleaved NrCAM serves as an excellent marker for monitoring substrate-selective ADAM10 activation *in vitro* and *in vivo* in patients.

**Impact**

This study sets the stage for a larger clinical phase III trial testing acitretin in AD. Moreover, it identifies a new companion diagnostic (NrCAM) for the clinical trials to monitor possible off-target effects of acitretin in patients and for developing additional, substrate-specific ADAM10-activating drugs.

analyzed with the software Maxquant (maxquant.org, Max-Planck Institute Munich) version 1.5.3.12. The MS data were searched against a reviewed fasta database of *Homo sapiens* from UniProt including isoforms (download: January 09, 2017, 42120 entries). Trypsin was defined as protease. Two missed cleavages were allowed for the database search. Carbamidomethylation of cysteine was defined as static modification. Acetylation of the proteins N-terminus and oxidation of methionine were set as variable modifications. The false discovery rate for both peptides and proteins was adjusted to < 1% using a target and decoy approach (concatenated forward/reverse database). Razor and unique peptides were used for quantification. Label-free quantification (LFQ) of proteins required at least two ratio counts of razor or unique peptides.

**Statistical analysis**

All tests were done on an exploratory 2-sided 5% significance level. *P*-values for Western blots were calculated with relative values using two-sided Student's *t*-test with a Welch's correction or one-way ANOVA with Dunnett's multiple comparison test for multiple hypothesis testing (software GraphPad Prism 7). Normal distribution was assumed.

For mass spectrometry, protein LFQ intensities after the treatment were divided by the related baseline values of the individual patient. The LFQ ratios (treatment/baseline) were log2-transformed, and a two-sided Student's *t*-test was applied to calculate significance levels for each protein between the acitretin-treated and the vehicle control groups. The sample size was chosen by the expected effect size of the respective experiments. No samples were excluded. The sample size of the CSF samples from human patients was chosen according to the availability of the material.

The CSF samples from patients treated with acitretin were analyzed blinded.

## Data availability

- Proteomics: The mass spectrometry proteomics data have been deposited to the ProteomeXchange Consortium via the PRIDE partner repository with the dataset identifier PXD010756.
- Imaging: The raw images of the neurite outgrowth experiment have been deposited to the Zenodo platform with https://doi.org/10.5281/zenodo.1344972.

Expanded View for this article is available online.

## Acknowledgements

We thank Anna Berghofer for excellent technical help with mass spectrometry sample preparation, Katrin Moschke for her excellent technical assistance with the quantitative real-time PCR, S. Reinhardt for experiments with rat neurons, and Carl P. Blobel for his helpful comments on the manuscript. This work was supported by the Deutsche Forschungsgemeinschaft (German Research Foundation) within the framework of the Munich Cluster for Systems Neurology (EXC 2145 SyNergy) and FOR2290, and by the Centers of Excellence in Neurodegeneration, and the Breuer Foundation Alzheimer award (to S.F.L.). T. Brummer was supported by a fellowship of the medical faculty of the Technische Universität München (Klinikum rechts der Isar).

## Author contributions

TB, FP-M, and SFL designed the research; TB, FP-M, and SAM analyzed the data; TB, SAM, and FY performed the research; TB and SFL wrote the paper; and FP-M, AF, TT, and KE contributed new important reagents or analytic tools.

## Conflict of interest

The authors declare that they have no conflict of interest.

## For more information

(i)   https://www.dzne.de/en/research/research-areas/fundamental-research/research groups/lichtenthaler
(ii)  Acitretin in AD: https://www.alzforum.org/therapeutics/acitretin
(iii) ADAM10 in AD: https://www.alzforum.org/alzpedia/adam10
(iv)  ADAM10 gene: https://www.ncbi.nlm.nih.gov/gene/102
(v)   ADAM10 protein: https://www.uniprot.org/uniprot/O14672
(vi)  NRCAM gene: https://www.ncbi.nlm.nih.gov/gene/4897
(vii) NrCAM protein: https://www.uniprot.org/uniprot/Q92823

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
