## [Review Process File · EMBO Molecular Medicine]

NrCAM is a marker for substrate-selective activation of ADAM10 in Alzheimer's disease

Tobias Brummer, Stephan A. Müller, Francisco Pan-Montojo, Fumiaki Yoshida, Andreas Fellgiebel, Taisuke Tomita, Kristina Endres and Stefan F. Lichtenthaler

Review timeline:	Submission date:	20 August 2018
	Editorial Decision:	9 October 2018
	Revision received:	8 January 2019
	Editorial Decision:	28 January 2019
	Revision received:	6 February 2019
	Accepted:	8 February 2019

Editor: Céline Carret

Transaction Report:

1st Editorial Decision

9 October 2018

Thank you for the submission of your manuscript to EMBO Molecular Medicine. We have now heard back from the three referees whom we asked to evaluate your manuscript.

You will see that the referees find the data interesting and convincing. Still, Referee 3 particularly points to a series of additional experiments that we believe would strengthen the conclusions of the paper and I would like to strongly encourage you to perform these.

We would welcome the submission of a revised version within three months for further consideration and would like to encourage you to address all the criticisms raised as suggested to improve conclusiveness and clarity. Please note that EMBO Molecular Medicine strongly supports a single round of revision and that, as acceptance or rejection of the manuscript will depend on another round of review, your responses should be as complete as possible.

I look forward to receiving your revised manuscript.

***** Reviewer's comments *****

Referee #1 (Comments on Novelty/Model System for Author):

I think this was a very careful piece of work. Think it is unlikely to have medical impact because I don't think a-sec potentiation is likely to be favoured therapeutic option (but that is only an opinion!!)

Referee #1 (Remarks for Author):

This is a careful but complicated piece of work which certainly suggests that actions of a-secretase are more complex than we had imagined in that up-regulation appears to differentially affect different substrates (one wonder if a-sec may be acting in different cellular compartments?). In the end, the paper is not a complete story (not meant as a criticism) because there is no explanation for this observation. But this is a nice piece of work which undoubtedly will lead to follow ups. The use of the NrCAM as a marker of activation (the papers title) is not quite ready for prime time because until one understands why there is differential effects, one worries that one could be misled. However... very nice and careful work. I have only one quibble... I don't think we can be confident that g-sec inhibition failed because of other g-sec substrates. I am sure as we look at b-sec inhibition and possibly a-sec activation we will revisit this issue. My own opinion (and it is only an opinion) is that APP/Ab is somehow the problem and not the other substrates

Referee #2 (Comments on Novelty/Model System for Author):

Paper represents a very important step forward in ADAM10 biology. The paper does a great job of showing that ADAM10 cleaves NrCAM both in vitro and in vivo, as well as cleaving ADAM10. They also show the importance of NrCAM in neuron biology. Finally the demonstration of differential effects of acitretin on ADAM10 substrates has important implications for the use of this agent for treatment of diseases such Alzheimer's disease.

Referee #2 (Remarks for Author):

Data clearly presented and adds significantly to our understanding of ADAM10 biology. Several small revisions would enhance the study.

1. q-RTPCR analysis of ADAM10 and NRCAM following acitretin treatment to see if it alters mRNA levels as this could help explain some of their findings -- this is mentioned but not done.
2. Legend of Fig 5 should indicate if the same gel is used for all samples -- the "smiling" of the mNrCAM would suggest not.
3. While Fig 1a is a cartoon -- modification so that moving the arrows for furin and ADAM10 to the approximate place the respective cleavage takes place would enhance reading.

Referee #3 (Remarks for Author):

In this manuscript Brummer and collaborators characterize the ADAM10-dependent cleavage of NrCAM. This shedding is important for neurite outgrowth and it is not regulated by acitretin, suggesting that substrate-selective ADAM10 activation may be feasible. Therefore, the NrCAM cleavage can be exploited for monitoring therapeutic ADAM10 activation towards APP.

The results could be relevant for the development of therapeutic strategies aimed at up-regulating ADAM10 activity selectively towards specific substrates, however I have two concerns.

First, the paper merges two main topics: (1) the characterization of ADAM10-dependent shedding of NrCAM and its role in neurodevelopment and (2) ADAM10-mediated-NrCAM cleavage as potential marker of selective ADAM10 activation towards APP in Alzheimer disease patients. The discussion of the two topics should be more coordinated.

Second, the authors show that acitretin affects sAPPalpha levels but not NrCAM shedding by ADAM10, however the mechanism underlying this effect is not reported. Acitretin increases ADAM10 transcription and protein levels, how does it influence the specificity of the enzyme? The authors describe the effect of TSPN15 in regulating the different cleavage of APP and NrCAM by ADAM10. However, a direct link between acitretin and TSP15 is not shown and it should be at least discussed in details.

Few points to address:

Fig.1: the authors analyse the levels of sNrCAM released in the medium using a N-terminal antibody. It would be important to check the specificity using also a C-terminal antibody for NrCAM, to verify that it does not recognize this band.

Fig.1 and FigEV1: sAPPalpha levels should be also analysed as positive control of the experiment.

Fig. EV2C

- According to the Western Blot analysis, the furin cleavage is not a limiting step of ADAM10 shedding of NrCAM, since in cultures treated with CMK the release of s-proNrCAM is detected. To have a clear picture of the process, a quantitative and statistical analysis of the results is required.
- the demonstration of the sequential cleavage of furin and ADAM10 should be confirmed using NrCAM mutants carrying either the furin or the ADAM10 mutated cleavage sites.
- In Fig. EV2C the authors should also include ADAM10 western blot, since furin affects also ADAM10 maturation, and sAPPalpha levels as control.

Fig. 3A, B

- The quantitative analysis of biotinylation assays should show the comparison of surface/total levels of the protein of interest. The authors show in fig.1B,C that treatment with ADAM10 inhibitor and deletion of the enzyme increase the levels of mNrCAM. Therefore, they should normalize the surface levels on the total levels to verify whether the increase in membrane expression of mNrCAM is related just to a general increase in the total levels of the protein.
- The authors should check the absence of an intracellular protein to demonstrate the reliability of the assay.
- The authors should check the mRNA levels of NrCaM in their experimental conditions.

Fig. 4 To strengthen the relevance of the selectivity of the acitretin-induced cleavage of ADAM10 substrates in Alzheimer Disease, the authors should perform the experiments shown in Fig4.A in mature neurons (DIV 12-14) and not in immature cells during development. Moreover, the authors should check also the cleavage of other neuronal substrates, as N-Cadherin and Neuroigin-1.

Fig.5A The authors should analyse also the levels of sAPPalpha as positive control of NMDA treatment.

Fig. 5B As requested for Fig. 3A,B, the quantitative analysis of biotinylation assays should show the comparison of surface/total levels of the protein of interest. Moreover, they should check the absence of an intracellular protein to demonstrate the reliability of the assay. Why ADAM10 membrane levels are not increased upon NMDA treatment?

Discussion: The authors cite that "PMA stimulated shedding of APP, but not NrCAM in primary murine neurons", however in the results section experiments showing PMA treatment are not presented.

Materials and methods: It is not clear how the authors analyse the Western Blot results: do they normalize the optical density on the optical density of a housekeeping gene?

Reviewer #1

This is a careful but complicated piece of work which certainly suggests that actions of a-secretase are more complex than we had imagined in that upregulation appears to differentially affect different substrates (one wonder if a-sec may be acting in different cellular compartments?). In the end, the paper is not a complete story (not meant as a criticism) because there is no explanation for this observation. But this is a nice piece of work which undoubtedly will lead to follow ups. The use of the NrCAM as a marker of activation (the papers title) is not quite ready for prime time because until one understands why there is differential effects, one worries that one could be misled. However... very nice and careful work.

I have only one quibble... I don't think we can be confident that g-sec inhibition failed because of other g-sec substrates. I am sure as we look at b-sec inhibition and possibly a-sec activation we will revisit this issue. My own opinion (and it is only an opinion) is that APP/Ab is somehow the problem and not the other substrates.

As suggested, we toned down the notion that g-secretase inhibitors were discontinued simply due to inhibition of Notch. The two new sentences simply state that g-secretase inhibitors were discontinued and that they did inhibit cleavage of multiple g-secretase substrates.

The new sentences in the introduction on top of page 4 now read as:” Thus, the intended activation of ADAM10 for a prevention or treatment of AD may induce mechanism-based side effects by interfering with the cleavage and physiological function of other ADAM10 substrates. While this has not yet been tested systematically, a precedent is seen for gamma-secretase inhibitors, which were discontinued in clinical trials for AD as they led to mechanism-based toxicity upon prolonged dosing (Golde et al, 2013). These inhibitors did not only block A β generation but also cleavage of additional gamma-secretase substrates, including Notch.

The new sentences in the discussion on page 16 now read as:” For example, gamma-secretase inhibitors, which have been discontinued for AD drug development, also block cleavage of numerous additional gamma-secretase substrates besides APP, including Notch (Golde et al, 2013). Thus, Notch cleavage is used to counter-screen gamma-secretase inhibitors to identify such compounds that spare Notch cleavage but still block APP cleavage (Golde et al, 2013).”

Reviewer #2

Paper represents a very important step forward in ADAM10 biology. The paper does a great job of showing that ADAM10 cleaves NrCAM both in vitro and in vivo, as well as cleaving ADAM10. They also show the importance of NrCAM in neuron biology. Finally the demonstration of differential effects of acitretin on ADAM10 substrates has important implications for the use of this agent for treatment of diseases such Alzheimer's disease. Data clearly presented and adds significantly to our understanding of ADAM10 biology. Several small revisions would enhance the study.

1. q-RTPCR analysis of ADAM10 and NRCAM following acitretin treatment to see if it alters mRNA levels as this could help explain some of their findings -- this is mentioned but not done.

As suggested, we conducted qPCRs for ADAM10 and NrCAM in acitretin-treated neurons. In agreement with previous publications (Holback et al, 2007; Koryakina et al, 2009; Tippmann et al, 2009), acitretin mildly, but significantly increased ADAM10 mRNA levels, while mRNA levels of NrCAM remained unaltered (Fig. EV4).

The new sentences in the results' section on page 8 are:” ADAM10 was also increased at the mRNA level (Fig. EV4A), as determined in the murine neurons and in line with previous studies (Endres et al, 2014; Tippmann et al, 2009). Surprisingly, however, acitretin did not increase sNrCAM levels (Fig. 4A and B). Moreover, NrCAM RNA levels as well as full-length NrCAM protein (proNrCAM and mNrCAM) were also not affected by acitretin treatment (Fig. 4A, B and Fig. EV4A).”

2. Legend of Fig 5 should indicate if the same gel is used for all samples -- the "smiling" of the mNrCAM would suggest not.

The “smiling” was only seen in the higher molecular weight part of that Western blot membrane. First, we blotted for NrCAM (high molecular weight range) on that particular membrane. Then, the membrane was cut to detect ADAM10 and actin in the lower half of the gel. A little “smiling” can still be seen for proADAM10 (Fig. 5A), which has a higher molecular weight than mADAM10. As suggested by the reviewer, we included a new sentence into figure legend 5:” The membrane was reprobed with the different indicated antibodies.”

3. While Fig 1a is a cartoon -- modification so that moving the arrows for furin and ADAM10 to the approximate place the respective cleavage takes place would enhance reading.

As suggested, the changes have been incorporated into Figure 1A. We also indicated the apparent molecular weight of the different NrCAM fragments.

Reviewer #3

In this manuscript Brummer and collaborators characterize the ADAM10-dependent cleavage of NrCAM. This shedding is important for neurite outgrowth and it is not regulated by acitretin, suggesting that substrate-selective ADAM10 activation may be feasible. Therefore, the NrCAM cleavage can be exploited for monitoring therapeutic ADAM10 activation towards APP.

The results could be relevant for the development of therapeutic strategies aimed at upregulating ADAM10 activity selectively towards specific substrates, however I have two concerns.

First, the paper merges two main topics: (1) the characterization of ADAM10-dependent shedding of NrCAM and its role in neurodevelopment and (2) ADAM10-mediated-NrCAM cleavage as potential marker of selective ADAM10 activation towards APP in Alzheimer disease patients. The discussion of the two topics should be more coordinated.

To address this point, we included a new paragraph on page 16 between the discussion of the ADAM10-cleaved sNrCAM as a biomarker and its use in AD clinical trials on the one hand and the discussion of how ADAM10 cleavage alters NrCAM function on the other hand.

The new paragraph is: " ..., sNrCAM may be used to identify additional drugs besides acitretin that preferentially activate ADAM10 cleavage of APP.

Our study demonstrates that acitretin did not alter NrCAM shedding *in vitro* or *in vivo*, making it unlikely that the acitretin-mediated increase in ADAM10 leads to mechanism-based side effects resulting specifically from altered NrCAM function. However, our study clearly demonstrates that other ways of modulating ADAM10 levels or specifically altering ADAM10 cleavage of NrCAM have major consequences for neurite outgrowth. ...".

Furthermore, at the end of the paragraph discussing how ADAM10 alters NrCAM function, we link this topic back to its relevance in AD clinical trials (bottom of page 17 and top of page 18):" Taken together, these results suggest that alterations of ADAM10 activity, and therefore of NrCAM shedding and surface levels, may be a means of controlling synaptic plasticity, by strengthening the contacts of highly active synapses, while making them less susceptible to Sema3F-induced spine retraction. Yet, given our finding that acitretin did not alter NrCAM shedding, it is likely that the physiological functions of NrCAM are not altered in clinical trials with acitretin."

Second, the authors show that acitretin affects sAPP α levels but not NrCAM shedding by ADAM10; however the mechanism underlying this effect is not reported. Acitretin increases ADAM10 transcription and protein levels, how does it influence the specificity of the enzyme? The authors describe the effect of TSPAN15 in regulating the different cleavage of APP and NrCAM by ADAM10. However, a direct link between acitretin and TSPAN15 is not shown and it should be at least discussed in details.

As suggested, we added a new paragraph to the discussion on page 15: "Whether TSPAN15 shows an altered expression in the AD patients that were treated with acitretin, is not known, as the brains of these individuals are not available. Yet, while our TSPAN15 experiment served to demonstrate that certain TSPANs can indeed alter ADAM10 cleavage of some substrates versus others, it is unlikely that the acitretin-stimulated, selective increase in sAPP α in the AD patients is directly related to altered TSPAN15 expression, because acitretin (increased sAPP α , no effect on sNrCAM) and TSPAN15 expression (reduced sAPP α and increased sNrCAM) had opposite effects on cleavage of APP and NrCAM. However, it is well possible that acitretin mediates its relatively specific effect through other members of the TSPAN C8 family, which comprises besides TSPAN15 also TSPANs 5, 10, 14, 17 and 33. Among them, especially TSPAN5 and 14 are highly expressed in particular cell types within the central nervous system (Matthews et al, 2017) and may therefore contribute to the differential acitretin effects on various ADAM10 substrates. In fact, by increasing intracellular atRA levels, acitretin can have an impact on the expression of many genes, not only ADAM10 (Lane & Bailey, 2005). However, generation of mice deficient in or transgenically expressing those TSPANs will be needed to better understand the role of the TSPANs in controlling the substrate selectivity of ADAM10. At this point it also appears possible that acitretin mediates its effects through proteins other than TSPANs. One precedent comes from the ADAM10-homolog ADAM17, which does not bind TSPANs, but another multi-pass transmembrane protein, iRhom1 or iRhom2. Recent studies revealed that this binary interaction is in fact a ternary interaction with the

soluble protein FRMD8/iTAP (Kunzel et al, 2018; Oikonomidi et al, 2018), which controls stability and activity of the ADAM17/iRhom complex. Thus, it appears possible, that additional proteins may also affect ADAM10 binding to TSPANs and thus contribute to substrate selectivity. Such proteins may be alternative targets for acitretin.”

Fig.1

The authors analyse the levels of sNrCAM released in the medium using a N-terminal antibody. It would be important to check the specificity using also a C-terminal antibody for NrCAM, to verify that it does not recognize this band.

As suggested, to check the specificity of the sNrCAM band, we analyzed the conditioned neuronal media, as well as the respective lysates with an antibody against the C-terminal (intracellular) part of NrCAM. The new data are included as new Fig. EV1A. As expected, the C-terminally binding antibody detected the full-length proNrCAM (above 150kDa), but not the furin-cleaved mNrCAM (at slightly below 150kDa), which lacks the C-terminal 60 kDa due to furin cleavage. In contrast, the N-terminally binding antibody detected both proNrCAM and mNrCAM. The specificity of the bands was demonstrated with their absence upon NrCAM knock-down in the same experiment in Fig. EV1A.

As described in our manuscript (due to furin cleavage), mNrCAM in the lysate comprises only the furin-cleaved ectodomain of NrCAM and thus has the same molecular weight as the secreted NrCAM (sNrCAM). Thus, the corresponding band in Fig. EV1A is labeled with s/mNrCAM. Next we tested whether the C-terminally binding antibody would detect sNrCAM in the medium. As expected, no band was recognized at slightly below 150kDa, while the N-terminally binding antibody clearly detected this band. This demonstrates, that the secreted NrCAM indeed lacks the C-terminus.

Of note, on this gel in Fig. EV1A we loaded same volumes of lysate and conditioned medium. Given the comparably higher concentration of mNrCAM in neuronal lysate than shed sNrCAM in the medium, the sNrCAM band intensity is lower than the intensity of mNrCAM in the lysate. We also added one sentence to the first paragraph of the results' section on page 5:” The protein bands were specific for sNrCAM and mNrCAM as demonstrated with shRNA knock-down experiments of NrCAM and using different antibodies (Fig. EV1A).”

Fig.1 and FigEV1

sAPPalpha levels should be also analysed as positive control of the experiment.

We carried out the suggested experiment and included the blots for sAPP α as a positive control in Fig. 1 and Fig. EV1B. As expected, ADAM10 inhibition by GI254023x or a knockout of the protease strongly reduced sAPP α , while a knockout of ADAM17 did not affect constitutive sAPP α shedding (Fig. EV1B), which is in line with our previous publication (Kuhn et al, 2010).

Fig. EV2C

According to the Western Blot analysis, the furin cleavage is not a limiting step of ADAM10 shedding of NrCAM, since in cultures treated with CMK the release of s-proNrCAM is detected. To have a clear picture of the process, a quantitative and statistical analysis of the results is required.

As suggested, we added quantifications of s-proNrCAM and, in addition, of total sNrCAM levels (= sNrCAM + s-proNrCAM) to figure EV2C. Total-sNrCAM remained unaltered, while s-proNrCAM was significantly increased upon furin inhibition (Fig. EV2C), in agreement with a previous publication (Susuki et al, 2012). The quantifications are in agreement with our previous conclusion that furin is not involved in NrCAM shedding.

The demonstration of the sequential cleavage of furin and ADAM10 should be confirmed using NrCAM mutants carrying either the furin or the ADAM10 mutated cleavage sites.

This experiment has been done in a previous study that we cited in the first version of our manuscript. Thus, we did not repeat this experiment. That study found NrCAM to undergo

metalloprotease-dependent cleavage (Susuki et al, 2012, see Figure S1 in that publication), but the identity of ADAM10 as the NrCAM protease was unknown in that study. A mutation of the furin cleavage site did not block NrCAM shedding, but shifted the secreted form from sNrCAM (150 kDa) to s-proNrCAM (170 kDa), in line with our results in Fig. EV2C. Yet, a deletion of the cleavage site within the membrane-proximal juxtamembrane region of NrCAM did abolish NrCAM cleavage, which again is in line with our results in Fig. EV2C.

Together with our study, these results suggest that NrCAM indeed undergoes a sequential cleavage cascade by furin, then ADAM10 and finally γ -secretase.

In Fig. EV2C the authors should also include ADAM10 western blot, since furin affects also ADAM10 maturation, and sAPP α levels as control.

We added the blots for sAPP α and ADAM10 as new Fig. EV2D. In agreement with previous publications (Anders et al, 2001; Lopez-Perez et al, 2001), dec-RVKR-CMK significantly decreased both mADAM10- and sAPP α levels (Fig. EV2D).

Regarding the reviewer's suggestion in this point and the two points above (all referring to Fig. EV2), we added the following paragraph on page 6 in the results' section: "In agreement with previous publications (Anders et al, 2001; Lopez-Perez et al, 2001), dec-RVKR-CMK decreased both mADAM10 and sAPP α levels (Fig. EV2D). Additionally, it decreased sNrCAM, which results from furin cleavage, and increased instead the cleavage of the non-furin-cleaved soluble form of NrCAM (s-proNrCAM). Thus, total sNrCAM (sNrCAM + s-proNrCAM) levels remained unaltered (Fig. EV2C), in agreement with a previous study (Susuki et al, 2012). This suggests that small changes in mADAM10 levels affect APP shedding more strongly than NrCAM shedding. Together with prior results, using NrCAM-mutants carrying the mutated furin cleavage site (Susuki et al, 2012); we conclude that under non-inhibited conditions NrCAM first undergoes cleavage by furin and subsequently by ADAM10."

Fig. 3A, B

The quantitative analysis of biotinylation assays should show the comparison of surface/total levels of the protein of interest. The authors show in fig. 1B, C that treatment with ADAM10 inhibitor and deletion of the enzyme increase the levels of mNrCAM. Therefore, they should normalize the surface levels on the total levels to verify whether the increase in membrane expression of mNrCAM is related just to a general increase in the total levels of the protein.

We normalized the surface levels of mNrCAM to its levels in the cell lysate. The abundance of mNrCAM increased in the total lysates to the same extent as on the neuronal cell surface (now shown in Fig. 3A, B). In fact, we assume that mNrCAM in the lysate consists mostly of cell surface and TGN-localized mNrCAM. During transport from the ER to the cell surface, proNrCAM (not yet furin-cleaved) is found in ER and Golgi. Processing by furin takes place in the TGN or within secretory vesicles and yields mNrCAM, which therefore is expected to be mostly localized to TGN, secretory vesicles, the plasma membrane and potentially after endocytosis to some extent to endosomes. Thus, it does not appear surprising that the increase in cell surface levels of mNrCAM correlates with the change of mNrCAM levels in the total lysate. Regarding ADAM10, it is not even clear where exactly it cleaves its substrates, but plasma membrane as well as TGN are the generally assumed cellular localizations, as also recently shown (Tan and Gleeson JBC 2018 in press). Thus, a loss of ADAM10 cleavage of NrCAM would increase full-length NrCAM levels both in the TGN and at the cell surface.

The authors should check the absence of an intracellular protein to demonstrate the reliability of the assay.

We tested the efficiency of our surface biotinylation assay by using 100 μ g of total protein for a streptavidin pull-down and running it next to 10 μ g of total lysate. Then we compared the abundance of the intracellular protein β -actin in the pull-down to the total lysate. Despite using 10 times more protein for the streptavidin pull-down, we could hardly detect β -actin in those samples, while it was highly abundant in total lysates (Fig. EV3A). In contrast and as expected, mADAM10 was strongly enriched after the streptavidin pull-down compared to the total lysates (Fig. EV3A). Together, these results show the efficacy of our biotinylation assay. We refer to this control experiment in the results' section on page 7: "Indeed, surface mNrCAM levels were increased by

about 50% in conditional ADAM10 knock-out neurons compared to control-transduced neurons, as established by cell-surface biotinylation (Fig. 3A and Fig. EV3A).”

The authors should check the mRNA levels of NrCAM in their experimental conditions.

In order to test, whether an inhibition of ADAM10 by GI254023X is not only influencing NrCAM’s ectodomain-shedding, but may in addition affect ADAM10 or NrCAM at the transcriptional level, we conducted qPCRs for both mRNAs. Both ADAM10 and NrCAM mRNAs remained unaltered by the GI254023X treatment (Fig. EV3D), which is in line with the conclusion that changes in NrCAM’s cellular and extracellular levels are caused directly by an inhibition of its sheddase. The new experiment is referred to in the results on page 7:” Importantly, ADAM10 inhibition did not alter mRNA levels of either ADAM10 or NrCAM (Fig. EV3D).”

Fig. 4

- 1. To strengthen the relevance of the selectivity of the acitretin-induced cleavage of ADAM10 substrates in Alzheimer Disease, the authors should perform the experiments shown in Fig4.A in mature neurons (DIV 12-14) and not in immature cells during development.**
- 2. Moreover, the authors should check also the cleavage of other neuronal substrates, as N-Cadherin and Neuroligin-1.**

1. We tried to conduct the proposed control experiment with aged neurons, but unfortunately, our murine cortical neurons did not survive cell culture until DIV12-14. Even a reduction of the cell concentration per well did not enhance the survival of the cells until DIV12-14. Instead, we can offer data from a experiment in DIV21 rat neurons, which was completed before the first submission of this manuscript and is now included as new Fig. EV4. Acitretin increased ADAM10 and sAPPalpha, which confirms our experiments using murine neurons at the earlier time point.
2. To determine whether acitretin increases shedding of other ADAM10 substrates besides APP, we went back to our murine experiment, where (in contrast to the rat experiment) enough sample volume was available for further analysis. We blotted for the established ADAM10 substrate MT4MMP (Kuhn et al, 2016), and found that its shedding remained unaffected by acitretin treatment, similar to sNrCAM (new data included into Figure 4A, B). In agreement with the human CSF analysis, this result demonstrates that acitretin has a relatively specific effect on stimulating sAPPalpha release while leaving shedding of most other ADAM10 substrates unaffected.

Fig.5A

The authors should analyse also the levels of sAPPalpha as positive control of NMDA treatment.

We tried to detect sAPPalpha, as suggested by the reviewer, but, unfortunately, detection of endogenous sAPP α was not possible in this experiment. Detection of endogenous sAPPalpha with the available antibodies requires in our hands that the medium of the murine neurons is cultured for many hours (typically 1 to 2 days) to allow enough sAPPalpha to be released for successful Western blot detection. Yet, the NMDA stimulation experiment was done with a total collection time of only 30 minutes. Thus, our blots for sAPPalpha were blank.

However, we can provide an alternative which also shows that, as requested by the reviewer, NMDA does indeed activate the NMDA receptor. As a control for the NMDA-experiment we used the competitive NMDA-receptor antagonist D-APV. D-APV completely blocked the NMDA-induced increase in NrCAM-shedding, demonstrating that the observed changes in NrCAM levels were truly caused by an activation of the NMDA-receptor (Fig. EV4B). This experiment is referred to on page 9 in the results:” As a control, the NMDA-induced increase in NrCAM cleavage was blocked with the specific NMDA receptor antagonist D-APV (Fig. EV4B).”

Fig. 5B

As requested for Fig. 3A,B, the quantitative analysis of biotinylation assays should show the comparison of surface/total levels of the protein of interest. Moreover, they should check the

absence of an intracellular protein to demonstrate the reliability of the assay. Why ADAM10 membrane levels are not increased upon NMDA treatment?

A. As suggested, we normalized the surface levels of mNrCAM to its levels in the cell lysate and observed a significant decrease of mNrCAM on the surface compared to the cell lysates, revealing that a rapid, short-term activation of NrCAM-shedding by NMDA strongly affects the cell surface pool of mNrCAM (Fig. 5B). In contrast, inhibition of ADAM10 (in Fig. 3), which was a long-term process, affected both pools of NrCAM – at the surface and within the cells.

B. The reliability of our surface biotinylation assay was tested and described above (see comment to Fig. 3A, B).

C. NMDA has previously been shown to drive ADAM10 to the synapse, but ADAM10 is also found outside of the synapse, such as along dendrites (Marcello et al, 2007). Thus, in line with the suggestion of the reviewer, we would expect that NMDA increases ADAM10 levels at the synapse. Yet, our cell surface biotinylation assay is not able to discriminate between the cell surface specifically at the synapse and the cell surface at other parts of a neuron, such as along dendrites. Thus, if only synaptic, but not extra-synaptic ADAM10 increases upon NDMA treatment, we would not expect a major increase in total surface ADAM10 levels, in line with our experimental result.

Discussion

The authors cite that "PMA stimulated shedding of APP, but not NrCAM in primary murine neurons", however in the results section experiments showing PMA treatment are not presented.

We are sorry for not having been clear enough in the first version of the manuscript, where we were referring to the established concept, that PMA increases sAPP α shedding. Thus, in Fig. EV1B, we had only shown for NrCAM that its shedding remains unaffected by a stimulation with PMA. Now, we followed the reviewer's suggestion and included as new Figure EV1C the experiment demonstrating that PMA does indeed increase release of sAPP α , but not of sNrCAM. We refer to the new figure EV1C at the end of the first paragraph of the results' section.

Materials and methods

It is not clear how the authors analyse the Western Blot results: do they normalize the optical density on the optical density of a housekeeping gene?

The protein concentration was measured using a BCA-assay and equal protein concentrations for every sample were then subjected to SDS-PAGE separation. Hence, the amount of sample per lane was normalized to the total protein concentration of each sample. 15-20 μ g of total protein was used for SDS-PAGE separation. In order to have a second line of control, we also blotted for a housekeeping gene (actin or calnexin), to assure the initial normalization was done properly. The western blots were quantified by Multi Gauge software (version 3.0). The values of the band intensity were normalized to the mean of the respective control values in each experiment. Both paragraphs above are now included in the methods' section on Western blotting on page 20.

2nd Editorial Decision

28 January 2019

Thank you for the submission of your revised manuscript to EMBO Molecular Medicine. We have now received the enclosed report. As you will see the reviewer is now supportive and I am pleased to inform you that we will be able to accept your manuscript pending minor editorial amendments.

***** Reviewer's comments *****

Referee #3 (Remarks for Author):

The authors have fully addressed my concerns.

Corresponding Author Name: Stefan F. Lichtenthaler

Journal Submitted to: EMBO molecular medicine

Manuscript Number: EMM-2018-09695-V2